# TRAINING-FREE MULTI-TOKEN PREDICTION VIA PROBING

## ABSTRACT

Large Language Models (LLMs) possess latent capabilities for multi-token prediction (MTP), despite being primarily trained for next-token generation. We introduce a simple, **novel, and training-free MTP** method that probes an LLM using **on-the-fly generated mask tokens** derived from the model's own embedding space. These mask tokens guide the model to predict multiple future tokens in parallel without modifying model weights or relying on external draft models. To assess the quality of these predictions, we propose a dynamic token tree construction mechanism based on cumulative token probabilities, coupled with a simple pruning algorithm that removes redundant token paths. Our method supports multiple mask token designs, enabling flexible probing strategies to improve parallel prediction quality. We also define block complexity—the number of tokens processed per model iteration—as a key setting for controlling compute usage during inference. Under equal block complexity, our approach consistently outperforms existing training-free baselines such as Lookahead decoding and Prompt-lookup-decoding in terms of **block efficiency** (acceptance rate), significantly **reducing number of model calls**. Finally, we provide **theoretical and empirical justification for why our method works**. We show that decoder layers progressively align mask token representations with next true token states, and formalize this connection through a lemma linking cosine similarity to Top-$K$ prediction accuracy, supported by empirical evidence. We evaluate our method on Spec-Bench using `LLaMA3` and `Qwen3` models, providing both quantitative and qualitative analyses. Our probing-based MTP improves block efficiency by $\sim 12\%$ for `LLaMA3` and 8–12% for `Qwen3` models and tokens per second by upto $\sim 15 - 19\%$ over baselines **without retraining or auxiliary models**.

## 1 INTRODUCTION

Recent work in LLM inference has explored **multi-token prediction (MTP)** as a way to better utilize GPU parallelism, reduce latency, and accelerate generation. Traditional autoregressive decoding generates one token per step, leaving significant compute underutilized. MTP methods aim to predict multiple future tokens in parallel (Gloeckle et al., 2024), but often rely on training auxiliary heads, modifying base model weights, or using external draft models, which are common in speculative decoding frameworks Cai et al. (2024); Chen et al. (2023); Leviathan et al. (2023).

However, training models—even small ones—requires substantial effort, including dataset generation, architecture tuning, and days of GPU compute (Cottier et al., 2024; Goel et al., 2024). Moreover, these methods introduce additional parameters and memory overhead ((Cai et al., 2024) introduces additional LM heads, where each head size is $\sim 400M$ for `LLaMA3.2-3B-Instruct` (Dubey et al., 2024)), making them less suitable for **edge devices** and compute-constrained environments. In contrast, training-free methods offer plug-and-play alternatives that work with frozen models, **require no retraining**, and **generalize across model families and use cases with lossless generation**.

In many practical deployment scenarios—such as mobile inference, real-time applications, or high-throughput serving (Davies et al., 2025)—compute constraints limit the number of tokens that can be processed per model iteration. To capture this constraint, we define block complexity (BC) as the total number of tokens processed in a single forward pass. BC is a controllable setting that governs

the trade-off between parallelism and compute cost. Designing MTP methods that perform well under varying BC settings is crucial for real-world applicability.

In this paper, we propose a training-free, plug-and-play MTP method that works with any LLM. Our approach is grounded in the simple yet powerful idea of **probing the model's internal generative** capacity using **on-the-fly generated mask tokens**. These tokens are computed from the model's embedding space and injected into the prompt to guide multi-token prediction. The resulting tokens are verified in parallel by the base model, enabling efficient and accurate generation without any additional training.

To structure the predicted tokens, we introduce a dynamic token tree expansion mechanism that adaptively grows token paths based on cumulative probabilities, avoiding manual tree configuration. We further propose a lightweight pruning algorithm that removes redundant tokens across tree depths, improving diversity. Our method supports multiple mask token initialization schemes, including mean of prompt embeddings and sampling from token embedding space, enabling flexible probing strategies. We show that prompt-embedding mean initialization consistently performs best across `LLaMA3` models. To ensure practical scalability, we implement an efficient attention mask and position index generation strategy, which significantly improves token throughput—achieving up to 26% higher token-rate over standard generation for `LLaMA3.1-8B-Instruct`.

We evaluate our method on Spec-Bench (Xia et al., 2024b), a diverse benchmark covering summarization, translation, reasoning, coding, and math tasks, using both `LLaMA3` and `Qwen3` (Yang et al., 2025) model families. Our method consistently outperforms existing training-free baselines such as Lookahead Decoding (Fu et al., 2024) and Prompt Lookup Decoding (Saxena, 2023), achieving higher block efficiency, fewer model forward calls, and better token-rate across tasks and configurations.

Finally, we perform both quantitative and qualitative analyses of token acceptance behavior, revealing how acceptance depends mask token design and task type. Our method shows strong performance across both open-ended (writing, roleplay, etc.) and closed-ended tasks (summarization, math, reasoning, etc.), and is especially well-suited for edge devices operating under compute constraints. We make the following contributions.

1. **Training-free multi-token prediction via probing**: We introduce a novel paradigm for multi-token prediction that leverages probing with mask tokens in the base model's embedding space, enabling generation without additional training or external draft models.

3. **Dynamic tree expansion for flexible decoding**: We propose a dynamic tree expansion mechanism that adaptively grows the decoding tree based on predicted tokens, removing the need for manually designed tree configurations and improving decoding efficiency.

4. **Efficient implementation for static tree decoding**: We design a GPU-friendly implementation for static tree attention mask and position index generation by caching and incrementally updating these components during decoding. This avoids recomputation and significantly improves throughput for fixed tree structures.

5. **Theoretical and empirical justification for our method**: We prove that alignment between mask token and next true token representations (measured via cosine similarity) guarantees inclusion of the correct token in the Top-$K$ prediction set (acceptance). We support this with empirical analysis of representation evolution across layers.

6. **Comprehensive evaluation on SpecBench**: We conduct extensive experiments on the SpecBench benchmark, demonstrating the effectiveness and generality of our approach across multiple model families and decoding scenarios.

## 2 BACKGROUND

We begin by formalizing the notation for a base autoregressive language model. Let $f_\theta(x_{1:t})$ denote the output logits of a frozen LLM with parameters $\theta$, given a prompt sequence $x_{1:t}$. The model is trained to predict the next token $x_{t+1}$ using: $x_{t+1} \sim \text{softmax}(f_\theta(x_{1:t}))$

To enable multi-token prediction without modifying the model, we inject **mask tokens** $m_1, m_2, \ldots m_k$ into the prompt. These tokens are computed dynamically using the model's own embedding space and appended to the prompt as: $x_{1:t}, m_1, m_2, \ldots m_k$. Each mask token $m_i$ is

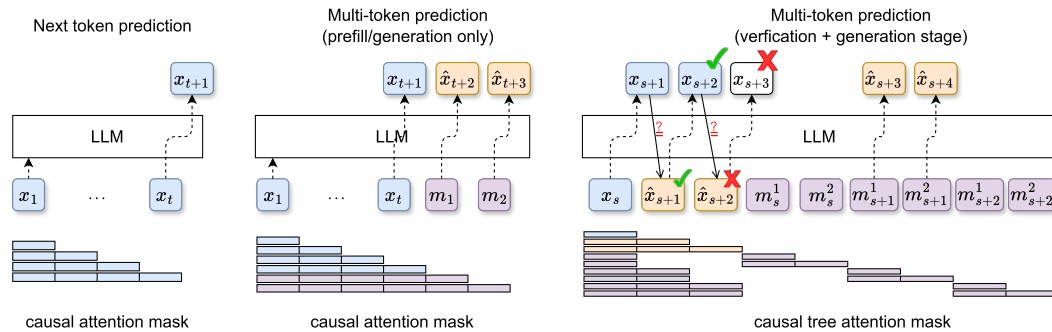

Figure 1: (Left) Standard next-token prediction setup for autoregressive models, (middle) multi-token prediction during prefill-stage by probing mask tokens which are appended to prompt tokens, (right) multi-token prediction with parallel verification and generation. Mask tokens are associated with last generated token ($x_s$) and future tokens ($\hat{x}_{s+1}, \hat{x}_{s+2}$) through custom tree attention mask.

designed to elicit a prediction of a future token $x_{t+1+i}$. The mask token pairs for last generated and predicted tokens—$m_{s+i}^1, m_{s+i}^2$ is same for all $i$—with unique conditioning based on causal tree attention mask to generate correct future tokens. The model processes the extended prompt and produces future tokens for each mask token position:

$$\hat{x}_{t+i+1} \sim \text{softmax}(f_\theta(x_{1:t}, m_1, m_2, \ldots, m_k)[t+1+i]) \tag{1}$$

To verify the predicted tokens, we perform simultaneous verification similar to (Lin et al., 2024) by appending each candidate token to the original prompt and checking whether the base model agrees with the prediction:

$$\hat{x}_{t+1+i} \stackrel{?}{=} x_{t+1+i} \sim \text{softmax}(f_\theta(x_{1:t+i})) \tag{2}$$

If the token $\hat{x}_{t+2}$ is accepted, it is appended to the prompt and used to verify the next predicted token $\hat{x}_{t+3}$ and so on. This sequential verification ensures that each accepted token is consistent with the model's own next-token prediction, guaranteeing same generation (lossless). This verification strategy is standard in speculative decoding and multi-token prediction literature Fu et al. (2024); Leviathan et al. (2023), but our method generated future tokens using mask token probing rather than draft model generation. We show detailed difference between vanilla LLM generation, LLM generation with mask tokens, and simultaneous verification and generation in Figure 1.

We defer the discussion of tree branching—where multiple candidate tokens are considered per mask token—to Section 3. There, we introduce a dynamic token tree construction mechanism that expands token paths based on cumulative probabilities and include pruning strategy to improve diversity.

## 3 METHODS

We propose a training-free multi-token prediction framework that probes frozen LLMs using dynamically generated mask tokens. These mask tokens are injected into the prompt and used to elicit predictions for multiple future tokens in a single forward pass. Our method doesn't require any auxiliary draft models or fine-tuning. The predicted tokens are verified sequentially using the base model itself, ensuring consistency with its autoregressive behavior. To support richer token exploration, we introduce a dynamic token tree construction mechanism that expands future token paths based on cumulative probabilities, along with a pruning strategy to eliminate redundant tokens. We also define block complexity as a key setting to trade-off parallelism and compute cost.

### 3.1 MASK TOKEN INJECTION

Let the input prompt be a sequence of tokens $x_{1:t} \triangleq [x_1, x_2, \ldots, x_t]$. These tokens are first projected into the embedding space via the model's embedding matrix $E \in \mathbb{R}^{V \times d}$, where $V$ is the vocabulary size and $d$ is the embedding dimension: $\mathbf{e}_i = E[x_i], \forall i = 1, \ldots, t$. To generate mask tokens, we explore several strategies:

a) **Prompt-based Hard Initialization**: Use the embeddings of last $k$ tokens, $m_i = \mathbf{e}_{t-k-i}, \forall i = 1, \ldots, k$

**b) Embedding Distribution based Initialization**: Let $\sigma$ be standard deviation over all embeddings and $\mu$ is mean over all embeddings of vocabulary size $V$, then sample each mask token embedding $m_i$ from Gaussian distribution, $m_i \sim \mathcal{N}(\mu, \sigma^2 I), \forall i = 1, \ldots, k,$ where, $\sigma = \sqrt{\frac{1}{V} \sum_{j=1}^{V} ||\mathbf{e}_j - \mu||^2}$

**c) Prompt Embedding Mean based Soft Initialization**: Compute the mean of prompt embedding for initial mask token, $m_i = \frac{1}{t} \sum_{i=1}^{t} \mathbf{e}_i, \forall i = 1, \ldots, k.$

During the generation phase, mask tokens are updated based on the tokens generated, adding more context-based information.

$$m_i[s+1] = m_i[s] + \lambda(\mathbf{e}_{t+s} - m_i[s]), \forall i \tag{3}$$

where $s$ denotes the generation step and $\lambda$ is a positive scalar.

We propose two prompt-context dependent strategies (re-initialized for new prompt) and one prompt-agnostic strategy for initializing mask token embeddings. These strategies allow us to probe the model using embeddings statistically similar to the prompt context, potentially revealing latent generative pathways. While the mask tokens take the same embedding value across all token trajectories in the future token tree, their position IDs and past context differ, leading to diverse generations.

## 3.2 WHY TRAINING-FREE MASK TOKENS ENABLE MULTI-TOKEN PREDICTION

Our method relies on the observation that decoder layers progressively enrich the mask token representation, aligning its hidden state with that of valid tokens. This alignment is critical because the LM head computes logits by taking inner products between the final hidden state and its vocabulary columns $W_r \in \mathbb{R}^d$. A higher inner product with a column corresponding to a valid token results in a higher logit, increasing the likelihood that the token appears in the top-K candidates.

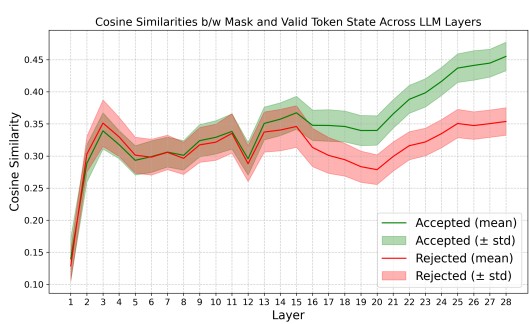

Figure 2: We use Dolly-Databricks (creative-writing) Conover et al. (2023) samples (100) to measure average cosine similarity across layers for mask and true-future token hidden-states. For `Llama3.2-3B-Instruct`, higher cosine similarity in later layers (15 onwards) correlates with token acceptance (green), while lower similarity correlates with rejection (red).

To quantify this alignment, we track the evolution of cosine similarity between mask and next true token hidden states across layers, specifically, for $n$ past tokens, we track hidden states of $x_{t+1}$ (next-true token), and $m_{t+1}$ (mask token), both trying to predict output token for position $t+2$ (next-next true token). As shown in Figure 3, accepted tokens exhibit a steady increase in cosine similarity after layer 15, reaching an average of about $0.45$, while rejected tokens plateau near $0.35$. This divergence suggests that higher similarity correlates with acceptance. We formalize this intuition with the following lemma:

**Lemma 3.1.** *Let $h_m, h_v \in \mathbb{R}^d$ be hidden states for the mask token and the next-true token after the last decoder layer and let $W \in \mathbb{R}^{d \times V}$ be the LM head with columns $w_r \in \mathbb{R}^d$. Assume $||h_m||_2, ||h_v||_2 \leq c_h$ and $||w_r||_2 \leq c_w, \forall r$. We define $i^* = \text{argmax}_r w_r^T h_v$ as the next-next true token (under greedy sampling) and $S_m = \text{argtopK}_r w_r^T h_m$ be the next top-K tokens under the mask token. Then,*

$$i^* \in S_m, \text{ if } cosine\_similarity(h_m, h_v) \geq \delta^* \text{ for some } \delta^* \in \mathbb{R}_{>0} \tag{4}$$

i.e., the next-next true token (under greedy sampling) belongs to the set of top-$K$ draft tokens generated from mask token when cosine similarity between next-true and mask token states exceed $\delta^*$ threshold. The proof is provided in Appendix in Section A.

## 3.3 LOGIT-BASED PREDICTION AND VERIFICATION

The output logit of each mask token can sample future tokens with conditioning on the past context. We sample Top-$K$ tokens from each mask token's logits to construct a tree of potential future

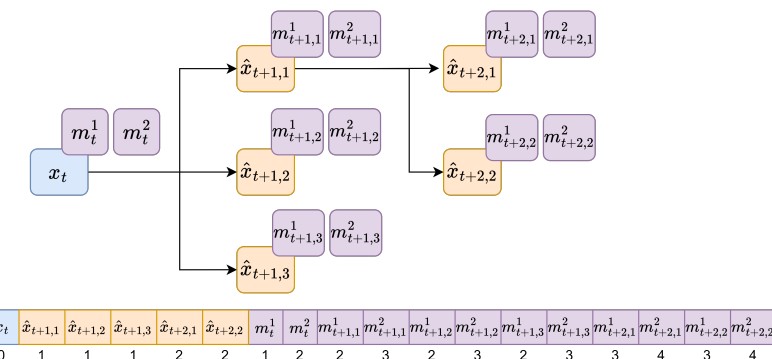

Figure 3: Mask tokens are present for each input token: last generated (blue) and future (orange) tokens, when processed by model all tokens are flattened and mask tokens are placed at the end with appropriate position indices.

tokens. At each depth, the Top-1 token is expanded with children tokens sampled from the next mask token. We prefer Top-1 expand to abide by block complexity bound and additionally favor high likelihood future token trajectories. We provide more detailed explanation of using Top-1 expand tree in Appendix Section D.

After the pre-fill stage, the generation stage includes both **verification and generation** as shown in Figure 1. The verification is performed by comparing each predicted token against the base model's next token output. A token is accepted if it matches exactly, ensuring **lossless generation**.

Once a particular token path is accepted, we pick the mask token(s) corresponding to the last accepted token for generating next set of future tokens. Each pair of mask token(s) is associated with last generated token and the future tokens having appropriate position index as shown in **??** and tree attention mask, shown in Figure 1 (right).

### 3.4 BLOCK COMPLEXITY (BC)

Training-free multi-token prediction methods often perform parallel verification and generation, increasing the number of tokens processed per forward model call which can become compute-bound in practical scenarios. We, therefore, define block complexity as the maximum number of input tokens processed parallely by the model in single pass. In our method for example, assume we use two mask tokens per future token and sample Top-$K_1$ and Top-$K_2$ tokens from first and second mask token respectively, then the block complexity is given by $3(1 + K_1 + K_2)$. The calculations are as follows:

1 input token for last generated token ($x_{t+1}$), $K_1 + K_2$ tokens for future tokens to be verified ($[[\hat{x}_{t+2,1:K_1}], [\hat{x}_{t+3,1:K_2}]]$), and lastly $2(1 + K_1 + K_2)$ mask tokens for each token trajectory ($[m_{t+1}^1, m_{t+2}^2, [m_{t+2,1:K_1}^1, m_{t+2,1:K_1}^2], [m_{t+3,1:K_2}^1, m_{t+3,1:K_2}^2]]$). Mask tokens are applied for each future token to be verified to enable parallel generation, and have different tokens to attend.

In general, for $k$ mask tokens and tokens sampled at each depth is: $K_i$, input complexity is:

$$\text{Block Complexity} = (k + 1)(1 + \sum_{i=1}^{k} K_i) \tag{5}$$

We compare baselines under matched block complexity, as higher complexity allows larger future token graphs and higher acceptance rates, but, can lead to higher latency and compute usage as well.

### 3.5 DYNAMIC TREE CONSTRUCTION

Constructing a tree of future token predictions typically requires selecting a fixed Top-$K$ from each mask token's logits. However, this approach is brittle and task-dependent, requiring tuning across models and domains. Instead, we propose a dynamic draft tree construction method that adapts to the model's uncertainty by using cumulative probability to determine best future token trajectories.

Our tree structure follows a **Top-1** expansion strategy, where only the highest-probability token is allowed to expand and form child nodes. This design simplifies the tree while preserving efficiency, as illustrated in Appendix Figure 6. We leave exploration of more complex tree structures to future

---

**Algorithm 1** Dynamic Generation of Token Tree

---

1: **Input:** LLM model $M$, Budget $B$, Mask token count $k$ and logits $l_{m_{1:k}}$, Input context $x$
2: **Output:** Draft token tree $T$ with $B$ nodes
3: Initialize tree $T$ with root node $r$, $P(r) \leftarrow 1.0$ ▷ Probability of root node is 1
4: nodes $\leftarrow \{r\}$, all_candidates $\leftarrow \emptyset$
5: **for** $i = 1$ to $k$ **do**
6:    new_candidates $\leftarrow \emptyset$
7:    **for all** node $n \in$ nodes at depth $i - 1$ **do**
8:        Get mask token logits $l_n$ from model $M$ given path of $n$
9:        Sample top-$(B - i)$ tokens $\{t_1, \ldots, t_{B-i}\}$ from $l_n$
10:       **for all** token $t_j$ in sampled tokens **do**
11:           Create child node $c$ with token $t_j$ appended to path of $n$
12:           $P(c) \leftarrow P(n) \cdot P(t_j | l_n)$ ▷ update cumulative probability
13:           Add $c$ to new_candidates
14:       **end for**
15:    **end for**
16:    all_candidates $\leftarrow$ all_candidates $\cup$ new_candidates
17:    **if** $i < k$ **then**
18:        Sort all_candidates by probability, take top $(B - i)$ as nodes
19:        all_candidates $\leftarrow$ all_candidates $\setminus$ nodes
20:    **end if**
21: **end for**
22: Add top $(B - 1)$ nodes from all_candidates to tree $T$
23: **Return** $T$

---

work. After getting all the token trajectories and cumulative probability, we chose the Top-$B - 1$, where $B$ is our block complexity and includes the last generated token. This allows the tree to grow adaptively—more branches are created when the model is uncertain, and fewer when it is confident.

Our algorithm, shown in Algorithm 1, takes as input the block complexity (budget) and the number of mask tokens (tree depth), and outputs a set of token trajectories that maximize coverage while respecting the computational budget. This avoids exhaustive grid search over tree branch (Top-$K$) and ensures that the tree structure is data-driven and model-aware. We show in Section 4 that dynamic tree generation performs on-par or better than hand-crafted tree branches.

### 3.6 TREE PRUNING

To reduce redundancy during tree expansion, we apply a simple pruning heuristic which removes consecutive repeated tokens—for example, when a child node predicts the same token as its parent (e.g., parent = "the", child = "the"). We observe that mask token predictions often include the last generated token or the parent token, which is typically redundant. To address this, we replace such token(s) with the next best token candidate from the mask token output distribution.

We perform ablation on using tree pruner and provide observations in Appendix Section G.5 that tree pruner helps improve avg token acceptance by up to $4\%$.

## 4 RESULTS

To evaluate the efficacy of our method, we conduct rigorous experiments using latest open-source frontier models: (a) `LLaMA3`, and (b) `Qwen3`, We use sample matching. where a token is accepted only if it is an exact match, enabling **lossless generation**.

**Models**: We evaluate two LLaMA3 models— `LLaMA3.2-3B-Instruct` and `LLaMA3.1-8B-Instruct`—and two Qwen3 models—`Qwen3-8B` and `Qwen3-32B`—to demonstrate that our method generalizes across architectures and scales. All models are run with a maximum generation length of 100 tokens on a single NVIDIA A100 GPU. The results below use greedy sampling, **results with temperature**=1.0 sampling are provided in the Appendix Section G.3.

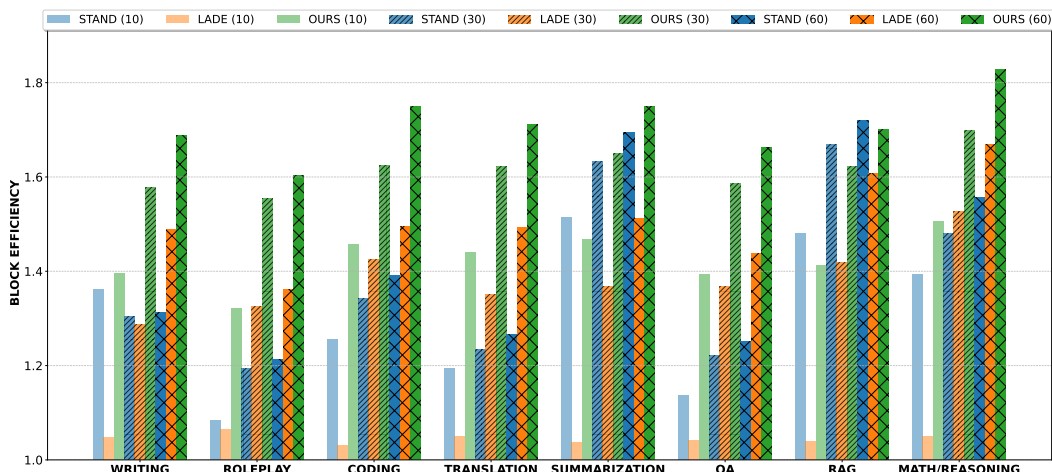

Figure 4: Evaluation on Spec-Bench using `LLaMA3.1-8B-Instruct` across block complexities (BC = 10, 30, 60). Our method consistently achieves the highest block efficiency across most tasks and BC settings.

**Tasks**: We use tasks from **SpecBench** Xia et al. (2024b), which includes summarization, translation, writing, coding, retrieval and math tasks (from GSM8K Cobbe et al. (2021)).

**Baselines**: We compare against training-free and draft-free speculative decoding methods: (i) **Prompt Lookup Decoding (PLD)** Saxena (2023), (ii) **Stochastic Adaptive N-gram Drafting (STAND)** Song et al. (2025), and (iii) **Lookahead Decoding (LADE)** Fu et al. (2024). The configuration for baseline methods is mentioned in Appendix Section F based on their respective papers.

**Performance Metric**: We report **Block Efficiency (BE)**, defined as the average number of tokens generated per model call: $BE = 1 + $ mean acceptance rate. BE directly reflects the reduction in model calls: model calls $\propto \frac{1}{BE}$, thus, **higher BE implies fewer model calls** and lower compute (energy) cost. We also report **tokens per second (T/S)** to show the absolute wall-time on A100 GPUs.

**Block Complexity (BC)**: We evaluate all methods at three block complexities: $10, 30, 60$.

**Mask token design** in our method is based on mean of given prompt's embedding (soft initialization) with dynamic updates based on the last token generated following Equation (3), with $\lambda = 0.1$. We use single mask token for BC=10,30 and two mask tokens for BC=60, unless otherwise stated.

Table 1: Comparison of multi-token prediction **block efficiency** (BE) and **tokens per second** (T/S) across models and methods averaged on Spec-bench tasks for block complexity $BC = 30$ and $BC = 60$

| Model | BC=30 | | | | | | | | BC=60 | | | | | | | |
|---|---|---|---|---|---|---|---|---|---|---|---|---|---|---|---|---|
| | PLD | | STAND | | LADE | | OURS | | PLD | | STAND | | LADE | | OURS | |
| | BE | T/S | BE | T/S | BE | T/S | BE | T/S | BE | T/S | BE | T/S | BE | T/S | BE | T/S |
| LLaMA3.2-3B-Instruct | 1.38 | 36.7 | 1.42 | 31.6 | 1.41 | 37.8 | **1.59** | **43.5** | 1.38 | 36.7 | 1.47 | 32.9 | 1.49 | 39.5 | **1.67** | **45.1** |
| LLaMA3.1-8B-Instruct | 1.30 | 32.7 | 1.38 | 29.2 | 1.38 | 32.6 | **1.62** | **38.9** | 1.30 | 32.7 | 1.43 | 31.0 | 1.51 | 35.6 | **1.71** | **40.5** |
| Qwen3-8B | 1.24 | 24.2 | 1.32 | 23.4 | 1.34 | 26.0 | **1.56** | **28.7** | 1.24 | 24.2 | 1.35 | 22.3 | 1.46 | 27.8 | **1.66** | **29.1** |
| Qwen3-32B | 1.27 | 13.1 | 1.32 | 13.1 | 1.33 | 14.9 | **1.54** | **15.9** | 1.27 | 13.1 | 1.37 | 14.2 | 1.45 | 16.1 | **1.65** | **16.3** |

We begin by reporting the **average block efficiency (BE)** of various methods on Spec-Bench tasks for two block complexities: **BC = 30 and BC = 60**. As shown in Table 1, our method consistently outperforms existing baselines, achieving up to **12%** **higher BE** on LLaMA3 models and **8–12% gains** on Qwen3 models over STAND, LADE which are SOTA training-free baselines. This translates to a substantial reduction in the number of forward model calls, up to **40% fewer model invocations** at BC = 30 and 60, as detailed in Appendix Section G.1 in Table 6. Notably, our method achieves these gains **without relying on any auxiliary N-gram cache**. Our method also gives best token rate, improving LADE by upto $\sim 15\%$ and $19\%$ for `LLaMA3.2-3B-Instruct` and `LLaMA3.1-8B-Instruct` respectively.

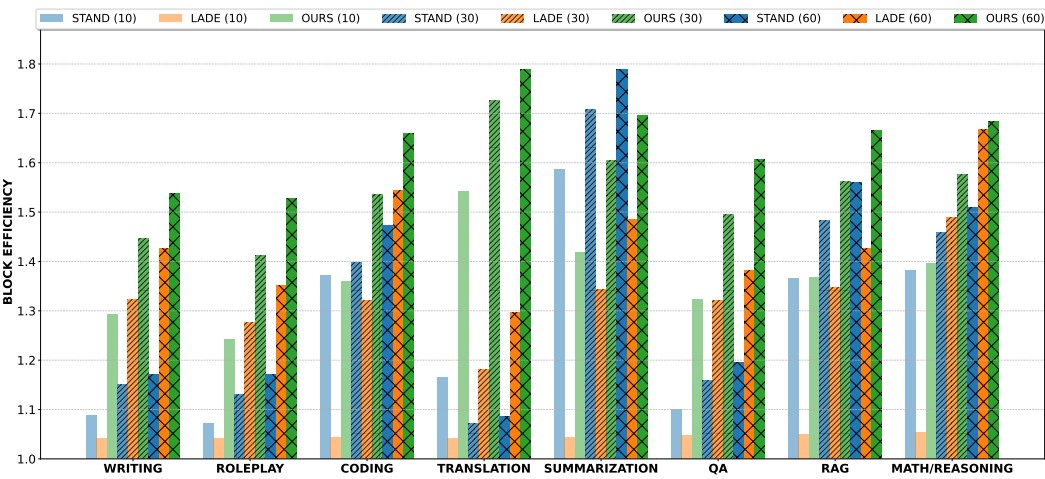

Figure 5: Evaluation on SpecBench using `Qwen3-32B` across block complexities (BC = 10, 30, 60). Our method consistently achieves the highest block efficiency across most tasks and BC settings.

We further present comprehensive BE results across all downstream tasks and block complexities (BC = 10, 30, 60) for `LLaMA3.1-8B-Instruct` and `Qwen3-32B`, shown in Figure 4 and Figure 5. Each method is color-coded, with higher opacity indicating larger block complexity. Our method (green) consistently achieves the highest BE across most tasks and BC settings, demonstrating that **LLMs, when probed effectively, can predict future tokens across diverse tasks**. For `LLaMA3.1-8B-Instruct`, LADE (orange) performs second-best across most tasks, except for 'summarization' and 'retrieval', where STAND (blue) shows stronger performance. Methods like STAND are particularly effective for these tasks, as a large portion of the generated tokens can be directly copied from the prompt. A similar trend is observed for `Qwen3-32B`.

For `LLaMA3.1-8B-Instruct`, the 'coding' task yields the highest gains in BE compared to other methods, while for `QWen3-32B`, the 'translation' task leads to higher gains compared to other baselines. Importantly, our method performs well even at **low BC**, making it suitable for **edge devices** where compute constraints limit block size. Exceptions include the 'retrieval' task on `LLaMA3.1-8B-Instruct`, where STAND slightly outperforms our method, and 'summarization' task on `QWen3-32B`, where our method ranks second, closely trailing STAND. Additional block efficiency results on `LLaMA3.2-3B-Instruct` and `Qwen3-8B` at BC = 60 are shown in Appendix Section G.2, Figure 8. Qualitative result of our method is shown in Appendix Section G.7.

> Key takeaway: for smaller block complexities (e.g., BC = 10, 30), probing with a single mask token surpasses existing baselines by a large margin. This empirically supports the hypothesis that LLMs, when probed appropriately, can confidently predict an additional token without requiring branching or memory overhead.

## 4.1 DYNAMIC TREE EXPANSION AND NUMBER OF MASK TOKENS TO PROBE

Table 2: Block-efficiency for block complexity 30 and 60 for `Llama3.2-3B`/`Llama-3.1-8B-Instruct` with one ($m_1$) and two mask ($m_1, m_2$) tokens across different branch configurations. The best BE is in bold while second best is underlined.

| Model | BC=30 | | | | | BC=60 | | | | | | |
|---|---|---|---|---|---|---|---|---|---|---|---|---|
| | $m_1$ | $m_1, m_2$ | | | | $m_1$ | $m_1, m_2$ | | | | | |
| | | [7,2] | [5,4] | [3,6] | dynamic | | [15,4] | [12,7] | [10,9] | [8,11] | [6,13] | dynamic |
| LLaMA3.2-3B-Instruct | **1.59** | 1.53 | 1.51 | 1.45 | 1.55 | **1.67** | 1.66 | 1.65 | 1.63 | 1.62 | 1.57 | **1.67** |
| LLaMA3.1-8B-Instruct | **1.61** | 1.55 | 1.53 | 1.47 | 1.57 | 1.708 | 1.696 | 1.69 | 1.67 | 1.65 | 1.60 | **1.712** |

To evaluate the effectiveness of our dynamic tree expansion, we perform an ablation study comparing different branching strategies under two probing setups: using a single mask token ($m_1$) and two mask tokens ($m_1, m_2$). Results for BC = 30 and BC = 60 are shown in Table 2. For BC = 30, the best performance is achieved with a single mask token, which does not require dynamic branching. In contrast, when using two mask tokens, dynamic branching consistently ranks either first or second, demonstrating its utility in larger tree configurations. For BC = 60, dynamic branching with two mask

tokens yields the better block efficiency across both `LLaMA3.2-3B`/`LLaMA3.1-8B-Instruct` models.

In Appendix Section G.4, we provide a comprehensive table comparing BE across tasks for both configurations. We observe that open-ended tasks (e.g., writing, reasoning) tend to perform better with a single mask token—benefiting from longer future token sequences and wider tree (greater exploration)—while closed-ended tasks (e.g., translation, math) perform better with two mask tokens deeper and less-wide tree (more focused and efficient exploitation of the model's predictions).

The size and structure of tree width (branches) are constrained by the number of mask tokens and the block complexity. For a single mask token, we use the maximum possible tree branch since only one future token is predicted. For example, with BC = 10 and 30, the tree branches are [4] and [14], respectively—no dynamic branching is needed in this case. For two mask tokens, multiple branching configurations are possible. For BC = 30, the tree can branch as $[9 - i, i]$ for $i \in 1, \ldots, 8$, allowing up to 9 branches. Similarly, for BC = 60, the tree can branch as $[19 - i, i]$ for $i \in 1, \ldots, 18$. Our dynamic tree expansion avoids exhaustive search over these configurations by adaptively selecting the optimal branching based on the input prompt, yielding strong BE performance.

Importantly, the number of tree branches tends to be smaller when more mask tokens are used, due to the need to respect the block complexity constraint. This trade-off can impact block efficiency, and our dynamic expansion strategy helps navigate it effectively.

## 4.2 EFFICIENT TREE ATTENTION MASK AND POSITION INDEX CONSTRUCTION

Tree-based decoding requires attention masks and position indices that respect branching hierarchies, which traditionally involves sequential iteration over tree nodes — a process that is **not GPU-friendly** and incurs high latency. To address this, we implement an **efficient strategy** that caches the attention mask and in-

Table 3: Token per second for ours (MTP) with **naive** and **efficient** implementation. Percentage improvements are relative to lookahead decoding.

| Model | Naive | Efficient |
|---|---|---|
| `LLaMA3.2-3B-Instruct` | 41.9 | 45.6 (**+15%**) |
| `LLaMA3.1-8B-Instruct` | 34.0 | 40.5 (**+14%**) |

crementally appends columns as new tokens are accepted, avoiding recomputation. Similarly, position indices are updated via a simple offset, enabling fast reuse across generation steps. These optimizations are detailed in Appendix Section E.

This approach significantly improves throughput for fixed tree structures. As shown in Table 3, our efficient implementation yields a 15% token-rate increase for `LLaMA3.1-8B-Instruct` and 14% for `LLaMA3.2-3B-Instruct`, compared to lookahead decoding. Gains are especially pronounced at higher block complexities, with improvements of 19–28% for `LLaMA3.1-8B-Instruct` and up to 20% for `LLaMA3.2-3B-Instruct` at BC = 60 (Table 4).

Table 4: Efficient inference implementation comparison for `LLaMA3.2-3B-Instruct` and `LLaMA3.1-8B-Instruct`. For two mask $(m_1, m_2)$ tokens static trees are used with branches $[7, 2]$ for BC=30 and $[15, 4]$ for BC=60.

| | LLaMA3.2-3B-Instruct | | | | LLaMA3.1-8B-Instruct | | | |
|---|---|---|---|---|---|---|---|---|
| **Method** | $m_1$(30) | $m_1, m_2$(30) | $m_1$(60) | $m_1, m_2$(60) | $m_1$(30) | $m_1, m_2$(30) | $m_1$(60) | $m_1, m_2$(60) |
| BE | 1.59 | 1.53 | 1.67 | 1.66 | 1.62 | 1.55 | 1.71 | 1.70 |
| Naive | 41.9 | 41.2 | 37.7 | 39.7 | 30.4 | 31.6 | 32.6 | 34.0 |
| Efficient | **43.5** | 42.3 | 45.1 | **45.6** | **38.9** | 38.2 | 39.9 | **40.5** |

## 4.3 DIFFERENT MASK EMBEDDING DESIGNS

We also evaluate different mask token initialization strategies, as described in Section 3.1. Among the variants tested, initializing the mask token using the **mean of the prompt embeddings** consistently yields the best performance across `LLaMA3` models, as shown in Table 5. This initialization provides a strong contextual prior, allowing the model to better align its predictions with the prompt semantics. These results suggest that even simple embedding-based heuristics can significantly influence multi-token prediction quality in training-free settings. We additionally run experiments to stress test the

Table 5: Different mask token initializations for `LLaMA3.2-3B-Instruct` and `LLaMA3.1-8B-Instruct`. Last K, Sample and Mean initialization are from Section 3.1

| Method | LLaMA3.2-3B-Instruct | | | LLaMA3.1-8B-Instruct | | |
|---|---|---|---|---|---|---|
| | $m_1(10)$ | $m_1(30)$ | $m_1, m_2(60)$ | $m_1(10)$ | $m_1(30)$ | $m_1, m_2(60)$ |
| Last K (hard init) | 1.36 | 1.53 | 1.62 | 1.38 | 1.56 | 1.67 |
| Sample (embedding distribution) | 1.39 | 1.57 | 1.65 | 1.41 | 1.60 | 1.69 |
| Mean (soft init) | **1.41** | **1.59** | **1.67** | **1.42** | **1.62** | **1.71** |

robustness of the mask token embedding design in Appendix Section G.6 and observed a minor performance drop when the mask token initialization is outside the embedding table distribution.

## 5 RELATED WORK

**Multi-token prediction (MTP)** accelerates LLM inference by predicting multiple tokens in parallel (Gloeckle et al., 2024; Guo et al., 2025). Recent methods include MEDUSA (Cai et al., 2024), which adds decoding heads and tree-based attention, and masked-input approaches with learnable sampler modules (Samragh et al., 2025). Other works train independent output heads atop a shared trunk (Gloeckle et al., 2024; Mehra et al., 2025) while finetuning the base model parameters. Unlike these, our method is training-free and operates on frozen LLMs using mask token probing and dynamic tree construction.

**Speculative Decoding (SD)** generates draft tokens for parallel verification (Leviathan et al., 2023; Chen et al., 2023). Extensions improve block efficiency via token trees (Miao et al., 2023; Sun et al., 2023). Some reuse target model layers for drafting (Zhang et al., 2023; Elhoushi et al., 2024), while others modify attention for multi-token prediction (Bhendawade et al., 2024; Lin et al., 2024). These approaches often require architectural changes or auxiliary models; ours does not.

**Training-free acceleration** includes Swift (Xia et al., 2024a), Jacobi decoding (Santilli et al., 2023), LADE (Fu et al., 2024), and PLD/STAND (Saxena, 2023; Song et al., 2025). These methods rely on caching or heuristic matching, whereas our approach avoids memory overhead and uses probing for dynamic tree construction.

Additional related work, using prompt/register tokens for MTP (Chen et al., 2024; Gerontopoulos et al., 2025), are discussed in Appendix Section B.

## 6 CONCLUSION

We present a training-free framework for multi-token prediction via probing, leveraging mask tokens from the embedding space to guide parallel generation. Our method introduces dynamic token tree expansion and block complexity-aware decoding, enabling efficient speculative inference without requiring draft models, N-gram caches, or offline token trees. Our theoretical insight reasons why probing with mask tokens leads to correct token prediction. Through extensive experiments across diverse tasks and model scales, we demonstrate that even a single mask token can yield substantial gains in block efficiency leading to reduction in model forward calls—often outperforming existing baselines—while dynamic branching with multiple mask tokens further enhances performance in deeper tree configurations. Our analysis reveals that open-ended tasks benefit from wider token trees, while closed-ended tasks prefer deeper, more focused trees. We also propose efficient implementations for attention masks and position IDs that significantly reduce runtime overhead. These findings collectively support the hypothesis that large language models, when probed appropriately, can confidently predict multiple future tokens without additional training or architectural changes.

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

# A  INTUITION FOR MULTI-TOKEN PREDICTION USING MASK TOKENS

This section provides the formal proof for the claim stated in Section 3.2 of the main paper, which involves comparing the hidden states of the next-true token and the forecast (mask) token when both attempt to predict the same position. As an example, for a prompt with $n$ tokens, we compare the hidden states of:

$$x_{t+1} \text{ (next-true token)}, \quad m_{t+1} \text{ (mask token)}$$

both trying to predict the $(t + 2)^{\text{th}}$ token:

$$x_{t+2}, \quad \hat{x}_{t+2,i} \text{ for } i \in \text{argtopK}_i \text{ logits}_{m_{t+1}}.$$

This analysis is only possible in hindsight, as during inference we do not have access to next-true token at the current generation step, and hope to generate next-next true using mask token.

We show theoretically that a higher cosine similarity between the hidden states of the mask token and the next-true token results in the next-next-true token being included in the Top-$K$ prediction set of the mask token. Intuitively, we believe that later decoder layers in LLMs enrich the mask token representation so that it aligns with the valid token state, thereby improving the likelihood of predicting correct next-next token.

**Lemma A.1.** *Let $h_m, h_v \in \mathbb{R}^d$ be hidden states for the mask token and the next-true token after the last decoder layer and let $W \in \mathbb{R}^{d \times V}$ be the LM head with columns $w_r \in \mathbb{R}^d$. Assume $||h_m||_2, ||h_v||_2 \leq c_h$ and $||w_r||_2 \leq c_w, \forall r$. We define $i^* = \text{argmax}_r w_r^T h_v$ as the next-next true token (under greedy sampling) and $S_m = \text{argtopK}_r w_r^T h_m$ be the next top-K tokens under the mask token. Then,*

$$i^* \in S_m, \text{ if } cosine\_similarity(h_m, h_v) \geq \delta^* \text{ for some } \delta^* \in \mathbb{R}_{>0} \tag{6}$$

*Proof.* For each $r$ we have:

$$|w_r^\top (h_m - h_v)| \leq ||w_r||_2 ||h_m - h_v||_2 \leq c_w ||h_m - h_v||_2 \tag{7}$$

Now compute:

$$||h_m - h_v||_2^2 = ||h_m||^2 + ||h_v||^2 - 2h_m^\top h_v.$$

Using $cosine\_similarity(h_m, h_v) \geq \delta$:

$$h_m^\top h_v \geq ||h_m|| ||h_v|| \delta \geq c_h^2 \delta.$$

So:

$$||h_m - h_v||_2^2 \leq 2c_h^2 - 2c_h^2 \delta = 2c_h^2 (1 - \delta).$$

Therefore:

$$||h_m - h_v||_2 \leq \sqrt{2} c_h \sqrt{1 - \delta}$$

Combining with equation 7 we get

$$|w_r^\top (h_m - h_v)| \leq c\sqrt{1-\delta}, \tag{8}$$

where $c = \sqrt{2}c_w c_h$. Now, define $\Delta_j^m = w_{i^*}^T h_m - w_j^T h_m$ and $\Delta_j^v = w_{i^*}^T h_v - w_j^T h_v$ be the difference in the logits between the token $i^*$ and $j$ under the mask token and next-true token respectively. We know that $\Delta_j^v \geq 0, \forall j$ since $i^* = \arg\max_r w_r^T h_v$. Also, using equation 8 we have:

$$w_{i^*}^T h_m - w_{i^*}^T h_v \geq -c\sqrt{1-\delta}$$
$$w_j^T h_v - w_j^T h_m \geq -c\sqrt{1-\delta}$$

Adding the above two inequalities we get:

$$\Delta_j^m - \Delta_j^v \geq -2c\sqrt{1-\delta}$$
$$\implies \Delta_j^m \geq \Delta_j^v - 2c\sqrt{1-\delta}$$

Hence, $\Delta_j^m > 0$ if $\Delta_j^v - 2c\sqrt{1-\delta} > 0$ which holds true when

$$\delta > 1 - \left(\frac{\Delta_j^v}{2c}\right)^2 \tag{9}$$

i.e., token $j$ has smaller logit value than token $i^*$ even under mask token output logits if Equation (9) holds.

Let $S_v \triangleq \text{argtopK}_r w_r^T h_v$ be the top-K tokens under the valid token $v$, and $K^*$ be the top-$K^{th}$ index for $w_r^T h_v$ under next-true token $v$ as well. Suppose $\delta > \delta^*$ where $\delta^* = 1 - \left(\frac{\Delta_{K^*}^v}{2c}\right)^2$, then we have the following:

$$\delta > 1 - \left(\frac{\Delta_{K^*}^v}{2c}\right)^2 \geq 1 - \left(\frac{\Delta_j^v}{2c}\right)^2, \forall j \notin S_v \tag{10}$$

This is because $\Delta_{K^*}^v \leq \Delta_j^v, \forall j \notin S_v$ due to the fact that the $K^*$ is in the top-K token set $S_v$.

Therefore using Equation (9), 10 we get,

$$\Delta_j^m = w_{i^*}^T h_m - w_j^T h_m \geq 0, \forall j \notin S_v$$
$$\implies w_{i^*}^T h_m \geq w_j^T h_m, \forall j \notin S_v$$

Hence, there are at least $V - K$ tokens for which $\Delta_j^m > 0$. Therefore, $i^*$ is in the top-K set $S_m$ when $\delta > \delta^* = 1 - \left(\frac{\Delta_{K^*}^v}{2c}\right)^2$

$$\square$$

Note that to match the top-1 next-next true token with mask token outputs, $\Delta_{i^*}^v = 0 \implies \delta^* = 1$, i.e., $\cos(h_m, h_v) = 1$. Thus, as the size of top-K set increases a smaller value of $\delta^*$ can still result in valid matches. To the best of our knowledge, this is the first proof to show connections between cosine similarity and acceptance in multi-token setting.

Finally, understanding why mask tokens achieve higher cosine similarity—specifically, how relevant information is injected into mask token states in the first place—remains an interesting question, which we leave for future work.

## B  EXTENDED RELATED WORKS

**Multi-token prediction** – Multi-token prediction (MTP) has emerged as a promising direction for accelerating LLM inference (Gloeckle et al., 2024) by leveraging parallelism and improving sample efficiency (Guo et al., 2025). MEDUSA (Cai et al., 2024) augments LLMs with multiple decoding heads and a tree-based attention mechanism to predict future tokens in parallel, while fine-tuning entire model. Samragh et al. (2025) introduce a masked-input formulation with gated LoRA adaptation and a learnable sampler module, enabling multi-token generation. Gloeckle et al. (2024) trains a model to predict multiple future tokens using independent output heads atop a shared trunk. Couple more methods (Chen et al., 2024), (Gerontopoulos et al., 2025) add learnable prompt/register tokens to base model to enable multi-token predictions, where similar idea has been proposed apriori in Lin et al. (2024). Chen et al. (2024) involves training prompt toens for each different model, and uses a static tree, whereas (Gerontopoulos et al., 2025) involves updating weights of the base model. Unlike these methods, our approach is entirely training-free and operates on any frozen LLMs, using mask token probing and dynamic tree construction to achieve efficient multi-token prediction without architectural changes or extra parameter overheads.

**Speculative Decoding** (SD) – SD accelerates LLM inference by generating draft tokens using a smaller or faster model, which are then verified by the target model in parallel (Leviathan et al., 2023; Chen et al., 2023). Several extensions improve block efficiency by generating token trees Miao et al. (2023); Sun et al. (2023); Jeon et al. (2024); Yang et al. (2024); Li et al. (2024). More recently, methods eliminate the need for a separate draft model by reusing target model layers. For example, Zhang et al. (2023) skips layers to derive a draft model with adaptive exit, while Elhoushi et al. (2024) uses early-exit verification trained with layer dropout. Bhendawade et al. (2024) replaces multi-head attention in the final layers with multi-stream attention to predict multiple tokens concurrently, requiring end-to-end training. Lin et al. (2024) introduces learnable mask and prompt tokens, enabling tree-based decoding and verification in a single forward pass. While effective, these methods often require architectural changes or auxiliary drafter models, unlike our approach, which is entirely training-free and drafter-free.

**Training-free Acceleration** – A growing body of work explores training-free approaches to accelerate autoregressive decoding. Swift (Xia et al., 2024a) proposes a training-free variant of Zhang et al. (2023) by selecting draft layers online via context-aware Bayesian optimization. Jacobi decoding (Santilli et al., 2023) formulates parallel decoding as a system of non-linear equations solved through fixed-point iteration, but consistently performs worst to LADE. LADE (Fu et al., 2024) tracks token trajectories using a fixed-size 2D window and generates n-grams in parallel, which are later verified by the target model. Prompt Lookup Decoding (PLD) (Saxena, 2023) matches the last generated n-gram to past context and, if matched, uses the subsequent token trajectory for verification. STAND (Song et al., 2025) extends PLD by storing logits of the entire past context and sampling a token tree from matched n-gram strings. Compared to LADE and STAND, our method avoids memory overhead from caching logits or token trajectories.

## C  FUTURE DIRECTIONS

**Mechanistic Interpretability**: Recent work in mechanistic interpretability (Olah, 2022) has focused on probing internal components of language models, such as linear layers and activation patching. While our probing-based multi-token prediction (MTP) method is not directly aligned with these approaches, we believe it offers a complementary perspective. Specifically, our use of mask tokens from the embedding space opens up new avenues for understanding LLM behavior. Future work could explore how different types of mask tokens interact with model internals, potentially revealing interpretable structures or activation patterns that guide multi-token generation.

**Combining Baseline Decoding Methods with MTP**: Our method can be synergistically combined with existing decoding strategies such as Lookahead Decoding (LADE) and Prompt-Lookup Decoding to further improve block efficiency and reduce the number of forward passes. One promising direction is to fuse token trajectories from both the N-gram cache in LADE and our probing-based method, then select top candidates from the combined pool. This hybrid approach could yield more robust and efficient decoding, especially in long-context or high-throughput settings.

**Efficient Dynamic Tree Attention Implementation**: In our current implementation, using multiple mask tokens with dynamic token trees results in varying attention masks across model passes. A naive solution is to precompute and store attention masks for all possible branch configurations, but this quickly becomes memory-intensive as block complexity (BC) increases. Future work could explore more efficient attention computation strategies, such as dynamic masking via sparse attention kernels or runtime mask synthesis, to reduce memory overhead while preserving flexibility. Additionally, integration with efficient inference frameworks such vLLM will also be explored, where tree-attention based speculative decoding method such as EAGLE-2 Li et al. (2024) are already integrated.

**Principled Design of Mask Tokens**: Our current work explores simple instantiations of mask tokens, but a more principled approach to mask token design could yield significant gains in block efficiency. For instance, control-theoretic or optimization-based methods to select mask configurations, may lead to more effective probing. Investigating the geometry and distribution of mask tokens in embedding space could also provide insights into their predictive power and generalization behavior.

**Combining with Diffusion LLMs (dLLMs)**: Recently, diffusion-based language models (dLLMs) have gained popularity for their competitive performance compared to autoregressive LLMs (AR-LLMs). These models are inherently trained using mask tokens with a bidirectional attention mask, a fundamental difference from the causal attention mask used in AR-LLMs. Several efficient designs have been proposed Sahoo et al. (2024); Wu et al. (2025); Israel et al. (2025). In future, it would be worth exploring designs that combine our method with bidirectional masked models capable of predicting multiple tokens by default.

## D  TOKEN TREE CONSTRUCT DETAILS

We perform **Top-1 tree expansion**, meaning that at each depth of the token tree, only the token with the highest probability is expanded to generate child nodes, following the approach in (Lin et al., 2024). This strategy becomes relevant when using two or more mask tokens.

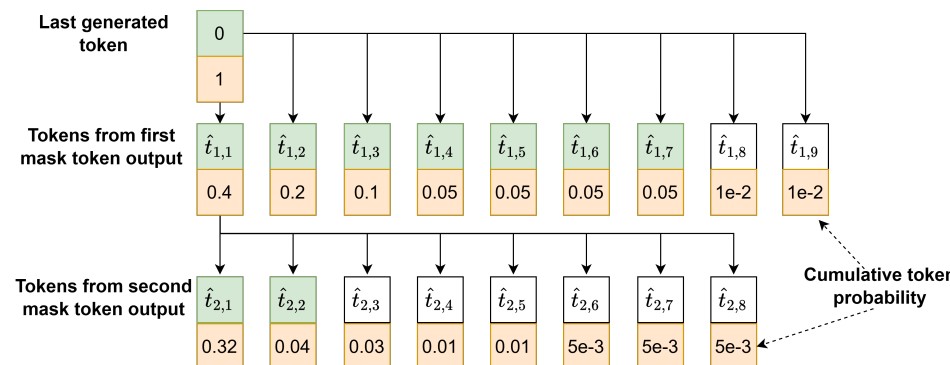

Figure 6: Example of dynamic token tree expansion for block complexity (BC) = 30 using two mask tokens. As the tree expands, each child node inherits the probability of its parent, resulting in a multiplicative score. We denote the last accepted token (root) as 0, the Top-9 tokens from the output logits of the first mask token as $\hat{t}_{1,1:9}$, and the Top-8 tokens from the second mask token as $\hat{t}_{2,1:8}$.

Our motivation for this choice stems from the observation that generations from mask tokens $(m_2, \ldots, m_k)$ exhibit **weak conditional dependence**. Let the output probability distribution of the LLM for input $x_{t-1}$ be defined as:

$$p_\theta(x_{t-1} \mid x_{<t-1}) \tag{11}$$

The output distributions from mask tokens are then:

$$p_\theta(m_1 \mid x_{t-1}, x_{<t-1}) \rightarrow \text{first mask token output} \tag{12}$$

$$p_\theta(m_2 \mid m_1, x_{t-1}, x_{<t-1}) \rightarrow \text{second mask token output} \tag{13}$$

$$p_\theta(m_k \mid m_{k-1}, \ldots, m_1, x_{t-1}, x_{<t-1}) \rightarrow k^{\text{th}} \text{ mask token output} \tag{14}$$

From the second mask token onward, the conditional generation includes previous mask tokens—serving as an approximation of actual token embeddings. As these mask tokens propagate through the model, their hidden states are refined, eventually predicting the correct token.

Expanding only the Top-1 token at each depth effectively **maximizes the approximate joint likelihood** of generation from multiple mask tokens. This expansion also helps maintain the token tree within a manageable block complexity (BC). Dynamic tree expansion is used to prioritize high-likelihood trajectories while avoiding exponential growth in tree size.

An example is shown in Figure 6 for two mask tokens ($m_1$, $m_2$) with BC = 30. In this case, we retain the Top-7 tokens from the output logits of $m_1$ and the Top-2 tokens from $m_2$, resulting in a compact yet expressive tree structure.

## E    EFFICIENT TREE ATTENTION MASK AND POSITION ID CONSTRUCTION

Using a token tree structure requires respecting its branching hierarchy during input construction, which introduces a sequential loop over tree nodes. This affects how position IDs and attention masks are computed—each must reflect the structure of the tree and the order of tokens in the input. In our proposed method, we found that custom tree attention mask generation is a sequential process that iterates over all branches of the token tree, resulting in high latency.

To address this, we exploit the static structure of the token tree—specifically when using a single mask token or two mask tokens with fixed branching—during the simultaneous generation and verification phase. As described in Section 3.4, for a given block complexity (BC), the input token order is fixed: last generated token, future predicted tokens, and mask tokens. Each of these components requires custom position indices (PID) and tree attention masks, as illustrated in Figure 1.

After the prefill phase, as generation progresses, both the PID tensor and attention mask evolve based on the number of tokens generated (i.e., block efficiency, BE) as shown in Figure 7. We observe that:

**Position IDs (PID)**: The only change across generation steps is a uniform shift in indices. The PID tensor from the previous step can be reused by simply adding the number of generated tokens (BE) to each index.

**Attention Mask**: The shape of the attention mask changes as the key-value cache grows. Specifically, the number of columns increases by BE, and these new columns are filled with zeros (assuming non-attended positions are represented by $-\infty$). This effectively prepends BE columns of zeros to the previous step's mask, yielding the correct attention mask for the current pass.

In Table 4 in main manuscript, we report token-rate improvements for token trees with varying number of mask tokens. For example, using a single mask token (with tree node counts of 15 and 30 for BC = 30 and 60, respectively), we observe: for `LLaMA3.2-3B-Instruct`, token-rate increases by $4\%$ for BC = 30 and $19.6\%$ for BC = 60. For `LLaMA3.1-8B-Instruct`, token-rate increases by $28\%$ for BC = 30 and $22.4\%$ for BC = 60. These results demonstrate that efficient reuse of attention masks and position IDs can significantly accelerate decoding, especially for larger token trees and larger models.

Note that, token-rate is influenced by both block efficiency (BE) and future token tree size, which depend on block complexity and the number of mask tokens. Our method scales well with tree depth and enables efficient decoding for high-capacity models. Future work will explore GPU-optimized dynamic tree expansion and integration with serving frameworks like vLLM (Kwon et al., 2023) and SGLang (Zheng et al., 2024).

## F    BASELINE CONFIGURATIONS

**PLD**: Uses a token tree of depth 10, with one token per depth. The future token trajectory is selected by matching the last generated n-gram to past context and verifying the subsequent tokens.

**STAND**: Constructs a token tree of depth 10, with additional branches at each depth based on block complexity. The number of tokens per depth is determined by selecting candidates with the highest cumulative probability from stored logits of matched n-gram strings. STAND is equivalent to PLD for smaller BC=10.

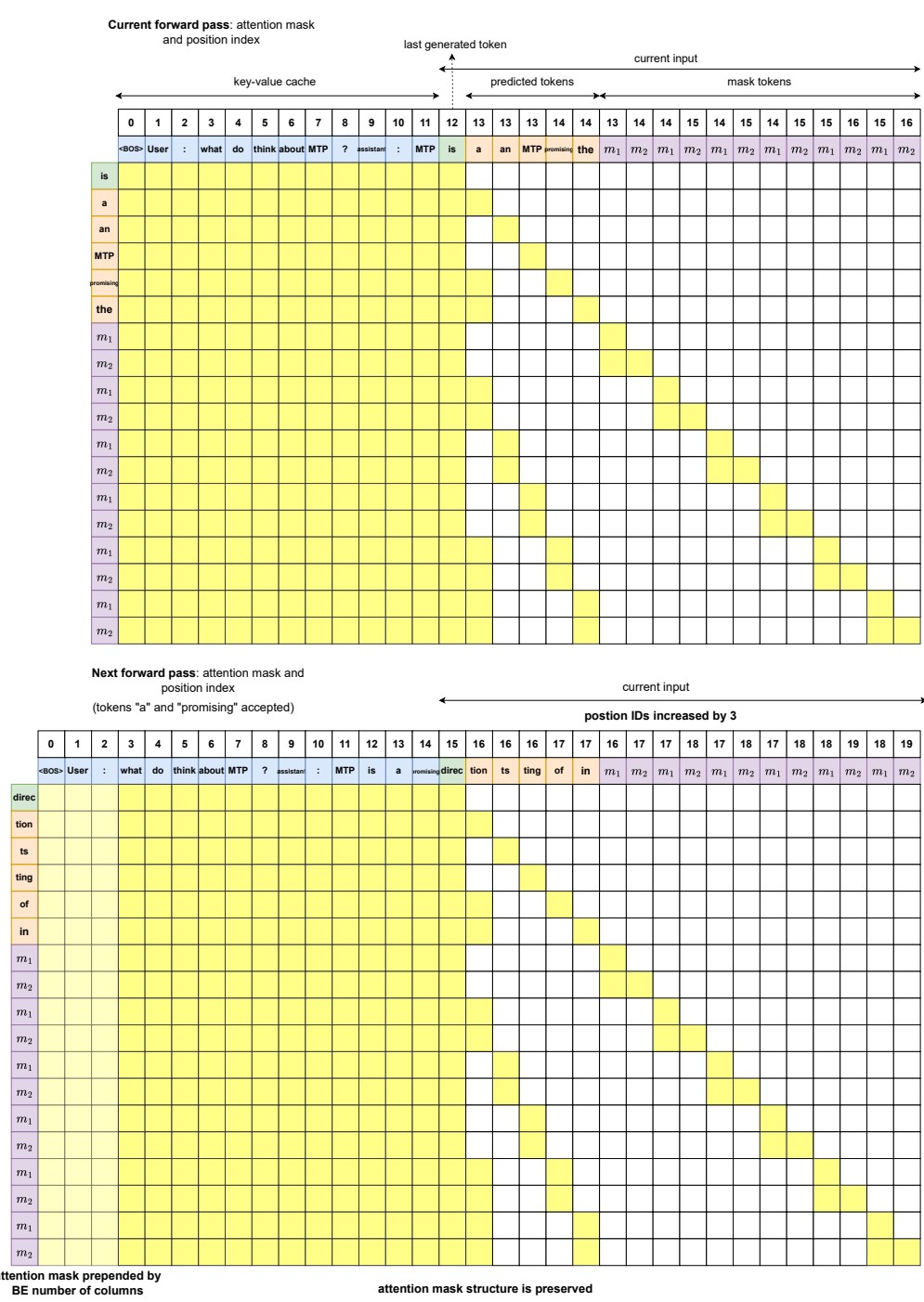

Figure 7: Attention mask and position index for the current (top) and next (bottom) LLM forward pass in our method with static branch configuration [3,2]. The tokens in blue are part of key-value cache, green token is the last generated token which will be inputted to the model to generate next token, orange tokens are future tokens, and purple tokens are mask tokens. Position IDs of each token is mentioned above it. The structure of the attention mask remains unchanged except for the addition of BE columns filled with zeros (assuming non-attended positions are represented by $-\infty$). Similarly, the position indices are preserved in order but uniformly incremented by BE.

**LADE**: Defines block complexity as $(L-1)(W+G)$

- $L$: N-gram size
- $W$: Window size
- $G$: Guess set size

Configurations used:

- BC=10: $L$=3, $W$=4, $G$=1 (minimum window size required is 3)
- BC=30: $L$=4, $W$=5, $G$=5
- BC=60: $L$=5, $W$=8, $G$=7

## G  ADDITIONAL RESULTS

We provide additional comprehensive results to offer deeper insights into the behavior and performance of the proposed method.

### G.1  REDUCTION IN MODEL FORWARD CALLS

Our method achieves the highest reduction in model forward passes, as shown in Table 6. Reducing the number of model calls helps save compute and energy, which is especially beneficial for edge or portable devices.

Table 6: Comparison of multi-token prediction **reduction in model forward calls** performance across models and methods averaged on Spec-bench tasks for block complexity $BC = 30$ and $BC = 60$

| Model | BC=30 | | | | BC=60 | | | |
|---|---|---|---|---|---|---|---|---|
| | **PLD** | **STAND** | **LADE** | **OURS** | **PLD** | **STAND** | **LADE** | **OURS** |
| LLaMA3.2-3B-Instruct | 27.54 | 29.58 | 29.08 | **37.11** | 27.54 | 31.97 | 32.89 | **40.12** |
| LLaMA3.1-8B-Instruct | 23.08 | 27.54 | 27.54 | **38.27** | 23.08 | 30.07 | 33.77 | **41.52** |
| Qwen3-8B | 19.35 | 24.24 | 25.37 | **35.90** | 19.35 | 25.93 | 31.51 | **39.76** |
| Qwen3-32B | 21.26 | 24.24 | 24.81 | **35.06** | 21.26 | 27.01 | 31.03 | **39.39** |

### G.2  BLOCK EFFICIENCY ON ADDITIONAL MODELS

We also compare BE across downstream tasks for `LLaMA3.2-3B-Instruct` and `Qwen3-8B` at BC = 60, as shown in Figure 8a and Figure 8b. Similar to results in Figure 4, 5, our method outperforms other baselines across most tasks, with the exception of 'retrieval' on `LLaMA3.2-3B-Instruct` and 'summarization' on `QWen3-8B`, where it performs second best. For `Qwen3` models, single mask token is used for BC=60 while for `LLaMA3` model two mask tokens are used.

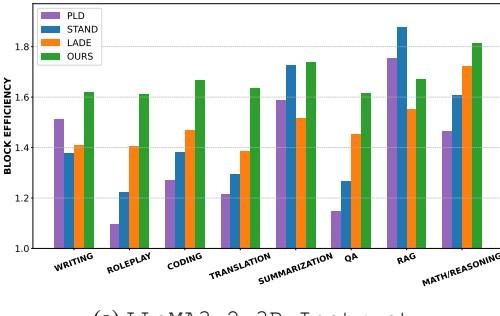 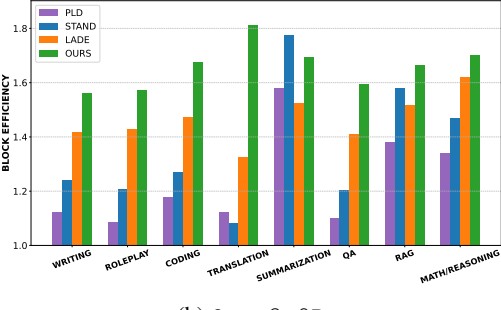

(a) `LLaMA3.2-3B-Instruct`         (b) `Qwen3-8B`

Figure 8: Block efficiency across SpecBench tasks for `LLaMA3.2-3B-Instruct` and `QWen3-8B` at block complexity (BC) = 60. Our method consistently outperforms baselines across most tasks, demonstrating strong performance in both open-ended and closed-ended settings.

## G.3 IMPACT OF SAMPLING

We run all methods with base model in sampling mode with **temperature=1.0** for BC=30, 60 and provide the block efficiency in for `Llama3.2-3B-Instruct` and `Llama3.1-8B-Instruct` in Table 7, 8 respectively. Our method outperforms other methods in terms of maximum number of mean accepted tokens on average across different SpecBench tasks showing the efficacy of our method for different sampling strategies.

Table 7: Block-efficiency for `Llama3.2-3B-Instruct` across SpecBench tasks for BC = 30 and BC = 60. Task acronyms: WRIT (writing), ROLE (roleplay), CODE (coding), TRANS (translation), SUMM (summarization), QA (question answering), RAG (retrieval-augmented generation), M/R (math/reasoning).

| Method | BC | WRIT | ROLE | CODE | TRANS | SUMM | QA | RAG | M/R | AVG |
|--------|-----|------|------|------|-------|------|------|------|------|------|
| PLD | 30 | 1.53 | 1.10 | 1.24 | 1.22 | 1.62 | 1.15 | 1.77 | 1.46 | 1.39 |
| STAND | 30 | 1.31 | 1.17 | 1.31 | 1.20 | 1.55 | 1.18 | 1.53 | 1.43 | 1.33 |
| LADE | 30 | 1.37 | 1.27 | 1.39 | 1.33 | 1.38 | 1.39 | 1.39 | 1.52 | 1.38 |
| OURS | 30 | 1.51 | 1.46 | 1.53 | 1.52 | 1.57 | 1.44 | 1.49 | 1.61 | **1.52** |
| STAND | 60 | 1.33 | 1.22 | 1.33 | 1.22 | 1.61 | 1.22 | 1.58 | 1.46 | 1.37 |
| LADE | 60 | 1.49 | 1.38 | 1.56 | 1.42 | 1.50 | 1.47 | 1.55 | 1.64 | 1.50 |
| OURS | 60 | 1.60 | 1.53 | 1.63 | 1.60 | 1.65 | 1.54 | 1.59 | 1.70 | **1.61** |

Table 8: Block-efficiency for `LLama3.1-8B-Instruct` across tasks for BC = 30 and BC = 60. Task acronyms: WRIT (writing), ROLE (roleplay), CODE (coding), TRANS (translation), SUMM (summarization), QA (question answering), RAG (retrieval-augmented generation), M/R (math/reasoning).

| Method | BC | WRIT | ROLE | CODE | TRANS | SUMM | QA | RAG | M/R | AVG |
|--------|-----|------|------|------|-------|------|------|------|------|------|
| PLD | 30 | 1.40 | 1.12 | 1.25 | 1.20 | 1.51 | 1.14 | 1.48 | 1.40 | 1.31 |
| STAND | 30 | 1.24 | 1.14 | 1.34 | 1.25 | 1.56 | 1.17 | 1.50 | 1.39 | 1.32 |
| LADE | 30 | 1.22 | 1.16 | 1.31 | 1.28 | 1.26 | 1.21 | 1.25 | 1.32 | 1.25 |
| OURS | 30 | 1.52 | 1.48 | 1.59 | 1.56 | 1.61 | 1.53 | 1.55 | 1.65 | **1.56** |
| STAND | 60 | 1.26 | 1.17 | 1.35 | 1.28 | 1.60 | 1.21 | 1.56 | 1.47 | 1.36 |
| LADE | 60 | 1.25 | 1.22 | 1.38 | 1.37 | 1.41 | 1.28 | 1.34 | 1.46 | 1.34 |
| OURS | 60 | 1.61 | 1.57 | 1.70 | 1.65 | 1.70 | 1.63 | 1.64 | 1.74 | **1.66** |

## G.4 IMPACT OF NUMBER OF MASK TOKENS

We observe that different downstream tasks benefit from different numbers of mask tokens, as shown in Table 10 for BC = 60. Tasks such as writing, roleplay, and question answering—more open-ended in nature—achieve higher block efficiency (BE) with a single mask (shallow and wider tree) token due to the larger number of predicted tokens. In contrast, tasks like coding, summarization, retrieval, and math/reasoning—more closed-ended—perform better with two mask tokens (deeper and focused tree), despite predicting fewer tokens.

More specifically, for BC = 60, probing with a single mask token $m_1$ results in a token tree size of 30 (shallow and wider), whereas using two mask tokens ($m_1$, $m_2$) yields a tree size of 20 (deeper and less wide).

For smaller BC = 30, using a single mask token consistently outperforms two mask tokens, as shown in Table 9. The two-mask configuration uses our dynamic tree expansion algorithm. We consider performance to be similar when the BE difference is within 0.01.

## G.5 IMPACT OF TREE-PRUNER

We evaluate the impact of our lightweight tree-pruner—which introduces no additional computational overhead—on block efficiency across SpecBench tasks for `LLaMA3.2-3B-Instruct` and `LLaMA3.1-8B-Instruct`, as shown in Table 11. The pruner is particularly effective when using

Table 9: Block-efficiency across different tasks for **BC=30** with different mask tokens probed (single mask: $m_1$, two mask tokens: $m_1, m_2$). Task acronyms are WRIT: writing, ROLE: roleplay, CODE: coding, TRANS: translation, SUMM: summarization, M/R: math/reasoning

| Model | Config | WRIT | ROLE | CODE | TRANS | SUMM | QA | RAG | M/R |
|-------|--------|------|------|------|-------|------|-----|-----|-----|
| `Llama3.2-3B-Instruct` | $m_1$ | 1.56 | 1.54 | 1.57 | 1.56 | 1.64 | 1.55 | 1.70 | 1.69 |
| | $m_1, m_2$ | 1.52 | 1.51 | 1.53 | 1.53 | 1.59 | 1.50 | 1.69 | 1.67 |
| `Llama3.1-8B-Instruct` | $m_1$ | 1.58 | 1.56 | 1.62 | 1.62 | 1.65 | 1.59 | 1.73 | 1.70 |
| | $m_1, m_2$ | 1.54 | 1.45 | 1.58 | 1.59 | 1.62 | 1.52 | 1.71 | 1.68 |

Table 10: Block-efficiency across different tasks for **BC=60** comparing different mask tokens probed. Task acronyms are WRIT: writing, ROLE: roleplay, CODE: coding, TRANS: translation, SUMM: summarization, M/R: math/reasoning

| Model | Config | WRIT | ROLE | CODE | TRANS | SUMM | QA | RAG | M/R |
|-------|--------|------|------|------|-------|------|-----|-----|-----|
| `Llama3.2-3B-Instruct` | $m_1$ | **1.64** | **1.65** | 1.64 | 1.65 | 1.72 | **1.64** | 1.78 | 1.78 |
| | $m_1, m_2$ | 1.62 | 1.61 | **1.67** | 1.64 | **1.74** | 1.62 | **1.83** | **1.81** |
| `Llama3.1-8B-Instruct` | $m_1$ | 1.67 | **1.64** | 1.73 | 1.70 | 1.74 | **1.69** | 1.80 | 1.78 |
| | $m_1, m_2$ | **1.69** | 1.60 | **1.75** | 1.71 | 1.75 | 1.66 | **1.87** | **1.83** |

two mask tokens, improving BE by up to $3\%$ for `LLaMA3.2-3B-Instruct` and up to $4\%$ for `LLaMA3.1-8B-Instruct`. For the single mask token setting, the pruner maintains BE without degradation.

This empirical result suggests that the **Top-$K$ predictions from the second mask token often include the Top-1 token from the first mask**, leading to redundancy. Our pruner removes such repeated tokens during tree expansion, enabling more diverse branching without sacrificing accuracy.

Table 11: Impact of tree pruning for `LLaMA3.2-3B-Instruct` and `LLaMA3.1-8B-Instruct`

| | `LLaMA3.2-3B-Instruct` | | | | `LLaMA3.1-8B-Instruct` | | | |
|--------|--------|--------|--------|--------|--------|--------|--------|--------|
| Method | $m_1(10)$ | $m_1(30)$ | $m_1, m_2(30)$ | $m_1, m_2(60)$ | $m_1(10)$ | $m_1(30)$ | $m_1, m_2(30)$ | $m_1, m_2(60)$ |
| w/o tree pruning | 1.41 | 1.59 | 1.50 | 1.63 | 1.42 | 1.62 | 1.51 | 1.66 |
| w/ tree pruning | 1.41 | 1.59 | **1.55** | **1.67** | 1.42 | 1.62 | **1.57** | **1.71** |

### G.6 IMPACT OF INITIALIZING MASK TOKENS OUTSIDE EMBEDDING DISTRIBUTION

To emphasize the robustness of mask token initialization, we compare soft initialization with sampling (as described in Section 3.1) against initializing the mask token using a high standard deviation ($\mu + 5\sigma, \mu + 10\sigma$), where $\mu$ and $\sigma$ denote the **mean and standard deviation of the entire embedding table**. We observe that block efficiency is only slightly affected, indicating that mask token initialization is robust across different settings. This further suggests that LLMs possess an inherent ability to predict future tokens.

Table 12: `Llama3.2-3B-Instruct` average block efficiency (BE) across Spec-bench categories with BC=60. Task acronyms are WRIT: writing, ROLE: roleplay, CODE: coding, TRANS: translation, SUMM: summarization, M/R: math/reasoning

| Method | WRIT | ROLE | CODE | TRANS | SUMM | QA | RAG | M/R | AVG |
|--------|------|------|------|-------|------|-----|-----|-----|-----|
| $\mu + \epsilon\sigma$ | 1.60 | 1.58 | 1.65 | 1.61 | 1.72 | 1.59 | 1.64 | 1.79 | **1.65** |
| $\mu + 5\sigma$ | 1.56 | 1.55 | 1.60 | 1.60 | 1.67 | 1.55 | 1.61 | 1.75 | **1.61** |
| $\mu + 10\sigma$ | 1.54 | 1.55 | 1.61 | 1.60 | 1.69 | 1.55 | 1.61 | 1.75 | **1.61** |

### G.7 QUALITATIVE EXAMPLES OF TOKEN ACCEPTED

To qualitatively evaluate the effectiveness of our probing-based multi-token prediction, we use the Dolly-Databricks dataset (Conover et al., 2023) focused on creative writing. This allows us to inspect

which future tokens are correctly predicted by the model without any training, simply through probing with mask tokens.

Below is a generation from `LLaMA3.2-3B-Instruct` when probed with two mask tokens under block complexity (BC) = 60. The model is able to predict several future tokens (blue) with just simple mask token probing—without any training or any N-gram cache.

**Prompt:**
`<|begin_of_text|><|start_header_id|>`user`<|end_header_id|>`
How do you make the best shepherds pie?
`<|eot_id|><|start_header_id|>`assistant`<|end_header_id|>`

Shepherd's pie, a classic British dish that's easy to make and always a crowd-pleaser! Here's a recipe for the best shepherd's pie, with a few tips and tricks to ensure it's a true culinary masterpiece:
**Ingredients:**
For the filling:

* 1 pound ground lamb or beef (or a combination of both)

* 1 onion, diced

* 2 cloves of garlic, minced

* 1 cup frozen peas and carrots

* 1 cup beef broth

* 2 tablespoons tomato paste

* 2 tablespoons all-purpose flour

* 2 teaspoons Worcestershire sauce

* 1 teaspoon dried thyme

* Salt and pepper, to taste

For the mashed potato topping:

* 3-4 large potatoes, peeled and chopped into 1-inch cubes

* 1/4 cup milk or half-and-half

* 2 tablespoons butter

* Salt and pepper, to taste

**Instructions:**
1.**Preheat your oven to 350°F (180°C).**
2.**Make the filling:**

* In a large pan over medium-high heat, cook the ground lamb or beef until browned,breaking it up witha spoon as it cooks.

* Add the diced onion and cook until softened, about 5 minutes.

* Add the minced garlic and cook for 1 minute.

* Stir inthe frozen peas and carrots, beef broth, tomato paste, flour, Worcestershire sauce, and thyme. Bring the mixture to a boil, then reducethe heat to low and simmer for 10-15 minutes, stirring occasionally.

* Season the filling with salt and pepper to taste.

3.**Make the mashed potato topping :**

* Boil the chopped potatoes in a large pot of salted water until they're tender, about 15-20 minutes.

* Drain the potatoes and return them to the pot. Add the milk, butter, salt, and pepper, an dmash the potatoes with a potato masher or a fork until they're smoothand creamy.

4.**Assemble the shepherd's pie:**

* Transfer the filling to a 9x13-inch baking dish.

* Spread the mashed potato topping evenly over the filling, making sure to cover the entire surface.

5.**Bake the shepherd's pie:**

* Place the dish in the preheated oven and bake for 25-30 minutes, or until golden and bubbling.

