# OpenReview forum: "Training-Free Multi-Token Prediction via Probing"
_ICLR.cc/2026/Conference — Submitted to ICLR 2026_

### Official Review · Reviewer_zinv · 2025-10-28

**Soundness:** 1
**Presentation:** 2
**Contribution:** 3
**Rating:** 4
**Confidence:** 4

**Summary:**

This paper presents a novel, training-free method for multi-token prediction (MTP). The core idea is to probe a large language model using mask tokens generated on-the-fly from the model's own embedding space, with the goal of significantly reducing the number of required model calls for generation. Another component of the proposed system is a dynamic tree expansion mechanism, which adaptively grows the decoding tree based on the predicted tokens. The authors validate their method through experiments on the SpecBench benchmark.

**Strengths:**

* The finding that an LLM's embedding space can be probed to elicit multi-token prediction is a surprising and interesting discovery.

* This finding leads to a novel, training-free approach that capably supports lossless generation while requiring fewer forward evaluations.

* The development of a dynamic tree construction mechanism with efficient caching represents a meaningful and practical systems-level contribution.

**Weaknesses:**

* The paper fails to discuss or contextualize itself against related work on masked diffusion LLMs, which represent another promising architecture for multi-token prediction using mask tokens. [1, 2, 3, 4]

* A significant weakness is the lack of explanation for why the proposed probing strategy works. The paper demonstrates that it works, but provides very little intuition or analysis regarding the underlying mechanism that makes this probing possible in a model trained exclusively for next-token prediction.

* The use of "block efficiency" as the primary metric is potentially misleading. While the method may decode 1.5 tokens per forward pass, this metric obfuscates the true wall-clock speedup. The overhead introduced by handling masked inputs and the verification process could diminish or even negate the gains. The authors should report throughput in their comparison against relevant baselines like STAND and LADE.

[1] Sahoo, Subham, et al. "Simple and effective masked diffusion language models." Advances in Neural Information Processing Systems 37 (2024): 130136-130184.

[2] Wu, Chengyue, et al. "Fast-dllm: Training-free acceleration of diffusion llm by enabling kv cache and parallel decoding." arXiv preprint arXiv:2505.22618 (2025).

[3] Hu, Zhanqiu, et al. "Accelerating diffusion language model inference via efficient kv caching and guided diffusion." arXiv preprint arXiv:2505.21467 (2025).

[4] Israel, Daniel, Guy Van den Broeck, and Aditya Grover. "Accelerating Diffusion LLMs via Adaptive Parallel Decoding." arXiv preprint arXiv:2506.00413 (2025).

**Questions:**

1. Given that the model is trained for next-token prediction, it is surprising that an input embedding can elicit a prediction at the position of the input mask rather than for the next position. Do the authors have any additional insight into the mechanism that enables this probing to work?

2. Speculative decoding can be extended to sampling [5], and most practical applications of LLMs rely on sampling to ensure response diversity. Is there a specific reason the experiments were limited to greedy decoding (as mentioned on line 269)?

3. For the sake of completeness, how does this training-free method compare in performance to existing approaches that require training?

[5] Chen, Charlie, et al. "Accelerating large language model decoding with speculative sampling." arXiv preprint arXiv:2302.01318 (2023).

---

> ### Author Response · Authors · 2025-11-20
> **Response to reviewer zinv**
>
> # General Response
> We thank the reviewer for recognizing the key insight of our work: probing LLMs with dynamically generated mask tokens enables accurate multi-token prediction without training. We believe this finding opens a new direction for understanding LLM internals.
>
> # Weakness 1: Lack of discussion on masked diffusion LLMs
> - We would like to mention that our focus is on autoregressive LLMs (AR-LLMs) and their ability to predict multiple tokens without training. Unlike dLLMs, which train mask tokens, use bidirectional attention and inherently perform multi-token prediction, our method uses causal attention and performs multi-token prediction in training-free manner **for models trained for only next-token prediction**.
> - We would like the reviewer to respectfully note that objective of dLLMs is to perform multi-token prediction trained using diffusion loss, whereas our work answers whether AR-LLMs trained for next-token prediction can exhibit multi-token prediction capabilities without architectural changes or retraining.
> - We will mention future work exploring connections between our method and dLLMs which could be interesting.
>
> # Weakness 2: Lack of explanation for why probing works
> - This is an excellent question and we have added intuition in the main paper (Section 3.2 (blue)).
> - At a high level, the mask token embedding can be viewed as a noisy version of the true token embedding. Decoder layers progressively “denoise” this representation, aligning it with the valid token’s hidden state.
> - Empirically, we show that cosine similarity between mask and valid token states increases across layers supporting this intuition. This increase is much higher for accepted tokens compared to rejected tokens as shown in Fig. 2.
> -  A formal proof relating cosine similarity to logit difference is provided in the Appendix section A.
>
> # Weakness 3: Block efficiency metric may obscure wall-clock speedup
> We agree and have added tokens/sec comparisons for all baselines and our method in Table 1 of the revised manuscript. Our method achieves higher throughput even when accounting for verification overhead.
>
> # Why only greedy decoding?
> - Most prior training-free and speculative decoding methods use greedy decoding for evaluation.
> - We have now added results for temperature = 1 decoding and observed that our method still outperforms baselines. Updated results are included in Table 7,8 in Appendix Section G.3.
>
> # Comparison with training-based methods
> - Training-based approaches (e.g., Eagle-2/3 [1]) often outperform training-free methods when trained on large datasets, but require **significant compute**, **retraining**, and **high quality dataset** for each new model.
> - Our method is plug-and-play across models without additional training, which we believe is a key practical advantage.
>
> We hope that these revisions and clarifications address the concerns raised and demonstrate the potential impact of our work. Based on these changes, we kindly request the reviewers to reconsider their evaluation.
>
> # References
> [1] Eagle-2 : Li, Yuhui, et al. "Eagle-2: Faster inference of language models with dynamic draft trees." arXiv preprint arXiv:2406.16858 (2024).

---

> ### Comment · Reviewer_zinv · 2025-11-24
> **Reviewer zinv Response**
>
> Thank you for your reponse. On W1 and W3, I am satisfied and I thank the authors for their clarifications and changes. I find the explanation given for W2 unsatisfactory, and I cannot reconsider my evaluation until it is adequately addressed.
>
> The relationship between cosine similarity and logit difference is straightforward because decoding a hidden state into logits is done with linear layer. That is not the core of my question and confusion.
>
> > At a high level, the mask token embedding can be viewed as a noisy version of the true token embedding. Decoder layers progressively “denoise” this representation, aligning it with the valid token’s hidden state.
>
> This explanation does not align with my understanding. LLMs are trained via next-token prediction loss, so it would be more accurate to say that decoder layers have been optimized to convert the representation of the i-th token, $h_i$, into the representation of the (i+1)-th token, $h_{i+1}$. Thus, if you input a masked token embedding $m_i$, I still expect it to output the embedding of the token that follows the masked token,  $h_{i+1}$, but instead it outputs the embedding $h_i$. This is my confusion: why isn't this shift a problem? I understand that a full mechanistic explanation may be impossible, but I hope the authors can at least provide some intuition as to how one may predict which method works best amongst a), b), and c) in Section 3.1, or even why these are natural approaches. Otherwise, the only explanation would be that one must try random methods and empirically check, which is deeply unsatisfying in a research paper.

---

> > ### Author Response · Authors · 2025-11-25
> > **Follow-up to reviewer zinv**
> >
> > We appreciate the reviewer’s insightful question and agree that a mechanistic explanation is challenging. However, we provide intuition grounded in model architecture and training dynamics.
> >
> > # Intuition for why shift isn’t a problem
> > LLMs trained on next-token prediction implicitly learn a strong autoregressive prior: each position’s representation encodes context and anticipates future tokens. When we insert a mask token embedding, the decoder treats it as a plausible continuation because self-attention distributes contextual information across all positions. Later layers refine this representation using surrounding tokens, effectively projecting the mask token into the manifold of valid token states. This progressive alignment explains why the mask token hidden state becomes closer to the next true token state, enabling accurate prediction of the next-next token.
> >
> > **New empirical evidence**: Figure 2 shows that for LLaMA3.2-3B-Instruct, cosine similarity between mask and next-true token hidden states increases after layer 15. Accepted tokens (green) exhibit a steeper increase compared to rejected tokens (red), supporting our alignment hypothesis.
> >
> > # More clarification on W2 using example
> > Let's consider that there are $n$ tokens generated, the mask token $m_{n+1}$ will have hidden state for any layer $l$: $$h_{m,n+1}^{l} = h_{x,n+1}^{l} + \epsilon^{l}$$
> > where $\epsilon^{l}$ is the perturbation/noise in layer $l$ compared to next-true token's hidden state. If standard NTP were used, we would first generate $x_{t+1}$ (next true token) and obtain hidden states $h_{x,n+1}^{l}$. Our claim is that the mask token hidden state aligns with this unknown true state sufficiently to predict the correct next-next token $x_{t+2}$ which will have the embedding $h^{0}_{n+2}$ ($l=0$ for embedding layer).
> >
> > # Why methods a), b), c) are natural
> > These designs are not random; they exploit different ways of initializing the mask token within the learned embedding space:
> >
> > - (a.)  Closest to prompt embedding: Leverages contextual similarity, making alignment easier.
> > - (b.)  Random within embedding distribution: Tests robustness and shows the model’s ability to recover from noise.
> > - (c.)  Average of previous tokens: Encourages smooth interpolation in embedding space.
> >
> > Our ablations (Table 5) shows that (c) always performs the best, and confirms that proximity to the embedding manifold correlates with better performance, supporting this intuition. Please note that table 5 was always part of the main paper.
> >
> > **New Table 12** in the Appendix shows that initializing mask tokens farther from the embedding manifold mean reduces performance, reinforcing that proximity to the learned space matters.
> >
> > # Connection to theory
> > Our updated Lemma (Section 3.2, Methods) formalizes this intuition: if the mask token representation is sufficiently aligned (cosine similarity $> \delta^{*}$) with next true token hidden state, the correct next-next token appears in the Top-K set.
> >
> > **We believe these clarifications, new empirical results, and theoretical guarantees fully address W2.**

---

### Official Review · Reviewer_7cXh · 2025-10-28

**Soundness:** 3
**Presentation:** 3
**Contribution:** 2
**Rating:** 4
**Confidence:** 5

**Summary:**

This paper proposes a training-free multi-token prediction (MTP) method that probes frozen large language models (LLMs) using dynamically generated mask tokens derived from the model’s own embedding space. These mask tokens enable parallel prediction of multiple future tokens without modifying model weights or using external draft models. The authors introduce a dynamic token tree construction mechanism guided by cumulative probabilities and a lightweight pruning strategy to improve prediction diversity and efficiency. Evaluated on Spec-Bench with LLaMA3 and Qwen3 models, their approach consistently outperforms existing training-free baselines in block efficiency, reducing model calls by up to 40% while maintaining lossless generation.

**Strengths:**

1. Training-free and plug-and-play: Requires no fine-tuning, auxiliary models, or architectural changes, making it easy to deploy across diverse LLMs.
2. Efficient dynamic decoding: The adaptive token tree expansion and pruning strategy improve block efficiency while respecting compute constraints (block complexity).
3. Strong empirical performance: Achieves consistent gains (8–12% higher block efficiency) over state-of-the-art training-free baselines across multiple tasks and model families.

**Weaknesses:**

1. A primary limitation is its practical deployability. The "plug-and-play" claim is misleading because the method's core relies on a highly custom "Causal Tree Attention Mask" for simultaneous verification and probing. This non-standard architecture is fundamentally incompatible with the highly-optimized, standard autoregressive mechanisms of SOTA inference frameworks like vLLM (e.g., PagedAttention). Integrating it would require a major re-engineering of the core attention and scheduling kernels, which contradicts the "plug-and-play" promise and is noted by the authors themselves as future work.

2. The method's generalizability is severely limited as all experiments are conducted only under greedy decoding (Temperature=0). This overlooks the predominant use of stochastic sampling (Temperature > 0, Top-P/K) in real-world applications like chatbots to ensure response diversity. The "probing" mechanism is fundamentally optimized to predict the highest-probability token path, which directly conflicts with the goal of sampling. Consequently, the moment a sampler selects a creative, non-greedy token, the entire probed draft tree is likely to be invalidated, causing the method's performance to collapse to that of standard autoregressive generation.

**Questions:**

- I understand that the authors use Block Efficiency to demonstrate the improvement in inference efficiency. However, I would also like to know about more conventional metrics, such as speedup ratio and mean number of accepted tokens per speculation step, as these would allow readers to make direct comparisons with other methods evaluated on SpecBench. I hope the authors could provide results using these standard metrics as well.

- Please provide the speedup results at temperature = 1, as measured by other speculative decoding (SPD) methods.

- Please adapt your method (e.g., by simplifying the tree structure) to integrate it with vLLM. I would like to see a more practical experimental result, and I hope the authors can provide the original throughput and the improved throughput.

I will adjust my score based on the author's Rebuttal answer

---

> ### Author Response · Authors · 2025-11-20
> **Response to Reviewer 7cXh**
>
> # General Response
> - We thank the reviewer for recognizing the strengths of our work, including its training-free nature, adaptive decoding, and strong empirical performance.
> - We would also like to highlight an additional novelty: the discovery that probing the model with dynamically generated mask tokens enables accurate prediction of future tokens. This phenomenon has not been demonstrated before and provides a foundation for understanding the inner workings of LLMs.
> - To further strengthen the paper, we have added a subsection in Methods (Section 3.2) explaining why mask tokens enable multi-token generation, supported by cosine similarity analysis across layers with theoretical justification.
>
> # Weakness 1: Practical deployability and plug-and-play claim
> - We appreciate the concern regarding integration with optimized inference frameworks. Our paper primarily addresses the research question:
>   - Can modern LLMs predict multiple tokens without additional training, and can this phenomenon be leveraged for speculative decoding?
>   - This is a fundamental capability question rather than an engineering optimization study.
> - Regarding integration with frameworks like vLLM:
>   - Eagle—a training-based speculative decoding method using tree attention masks—has official integration with vLLM [1].
>   - Our approach, which uses a simpler tree structure and avoids training, can similarly be integrated with vLLM with some/minimal engineering effort.
> - While benchmarking on vLLM is valuable for deployment scenarios, it is outside the scope of this work. We will clarify this distinction in the paper and mention integration as future work.
> - Note that other tree-based methods such as STAND and LADE also require tree attention masks and would need similar engineering effort.
>
> # Weakness 2: Generalizability beyond greedy decoding
> - Most training-free multi-token prediction and speculative decoding methods use greedy decoding as the standard evaluation setting.
> - Even under non-greedy sampling, the draft tree remains valid because it contains multiple branches, as shown in Figure 3, 6.
> - We ran all baselines with temperature = 1 and observed that our method still outperforms them. Updated results are included in Appendix in Table 7, 8 of the revised manuscript.
>
> # Metrics beyond block efficiency:
> - Block efficiency = 1+mean accepted tokens per speculation step. This metric is widely used in prior work, which is why we adopted it, as also done in [2], [3].
> - We have now added tokens/sec comparisons in Table 1 for direct comparison with other methods, our method outperforms all methods.
>
> We hope that these revisions and clarifications address the concerns raised and demonstrate the potential impact of our work. Based on these changes, we kindly request the reviewers to reconsider their evaluation.
>
> # References
> [1] Eagle integration with vLLM: https://docs.vllm.ai/en/v0.8.1/getting_started/examples/eagle.html
>
> [2] Goel, Raghavv, et al. "Direct alignment of draft model for speculative decoding with chat-fine-tuned llms." arXiv preprint arXiv:2403.00858 (2024).
>
> [3] Gagrani, Mukul, et al. "On speculative decoding for multimodal large language models." Proceedings of the IEEE/CVF Conference on Computer Vision and Pattern Recognition. 2024.

---

### Official Review · Reviewer_BqH7 · 2025-10-28

**Soundness:** 3
**Presentation:** 3
**Contribution:** 2
**Rating:** 2
**Confidence:** 5

**Summary:**

This paper proposes a training-free MTP method that probes an LLM using on-the-fly generated mask tokens derived from the model’s own embedding space. These mask tokens guide the model to predict multiple future tokens in parallel without modifying model weights or relying on external draft models. They also propose a dynamic token tree to assess the quality of these predictions, and use block efficiency to control compute usage during inference. Experimental results show the effectiveness of the method.

**Strengths:**

1. This paper is technically sound and easy to understand.
2. The experimental results show the effectiveness of the proposed method.

**Weaknesses:**

1. Why do you use block complexity/efficiency rather than tokens/s or mean accept tokens to judge the compute usage during inference when comparing to other training-free methods? It is more reliable to see if the actual speed-up of the proposed method is better than other competitors.
2. It is far from enough to show the token per second results of your model and base model in Tab.3. It is also far from enough to only validate results on two models. Since it is a training free method, it should be easy to generalize to many different types (Qwen, Deepseek, etc.) and sizes of models (70B). The authors should conduct experiments on more of them.
3. The paper focuses on generating tree structure and improve the overall MAT to speedup the large language model. However, they conduct experiments without using popular inference frameworks. Here comes a problem that the method may not have such speedup on the popular inference framework such as vLLM. In fact, HuggingFace Transformers framework does not optimize the speed of LLMs very well, which makes the ratio of the latency of tree generation process smaller. When using vLLM framework where the operations in LLMs are optimized very well, the tree generation process will take more time and reduce the speedup. The author should verify their method on such inference frameworks to show that their method is actually useful in reality.

**Questions:**

See weaknesses above.

---

> ### Author Response · Authors · 2025-11-20
> **Response to Reviewer BqH7**
>
> # General Response
> We thank the reviewer for acknowledging the technical soundness and clarity of our paper. We would like to reiterate that our key discovery is that autoregressive LLMs trained for next-token prediction inherently exhibit the ability to generate multiple tokens—a phenomenon we believe is novel and fundamental.
>
> # Weakness 1: Why block complexity/efficiency instead of tokens/sec?
> Block efficiency (BE) is directly related to mean accepted tokens (MAT):
> $$ BE =1+MAT$$
> as noted in Section 4 (lines 300–301). This metric is widely used in recent works such as TVD++ [1], SPD for MLLMs [2].
> - Conceptually, all tokens processed by the model can be viewed as a block, hence the term.
> - We also introduce block complexity, which becomes relevant for edge devices where large block sizes can reduce throughput. - Block complexity measures the total tokens processed (including extra tokens like masks or drafts).
> - To address the reviewer’s concern, we have added tokens/sec for all methods in Table 1 of the revised manuscript. **Our method consistently achieves higher tokens/sec compared to baselines**.
>
> # Weakness 2: Limited model coverage
> - We already evaluate on Qwen3 (latest Qwen series) and multiple sizes: 3B, 7B, 8B, and 32B, covering diverse families widely used in academia.
> - While we agree that testing on even larger models (e.g., 70B) would be ideal, the current set demonstrates generalization across families and scales, which we believe suffices for proof of concept. Many recent works [3], [4] only ablate on single model families.
>
> # Weakness 3: Lack of evaluation on optimized inference frameworks (e.g., vLLM)
> - Our paper primarily addresses the research question:
> *Can modern LLMs predict multiple tokens without additional training, and can this phenomenon be leveraged for speculative decoding?*
> - This is a fundamental capability question rather than an engineering optimization study.
> - Regarding integration with optimized frameworks such as vLLM:
>   - Eagle—a training-based speculative decoding method using tree attention masks—has official integration with vLLM [5].
> - Our approach, which uses a simpler tree structure and avoids training, can similarly be integrated with vLLM with engineering effort.
> - While benchmarking on vLLM is valuable for deployment scenarios, it is outside the scope of this work. We will clarify this distinction in the paper and mention integration as future work.
> - Additionally, note that lookahead decoding (LADE) and stochastic N-gram adaptive decoding (STAND) also uses tree attention masks, and therefore, tree attention mask is not unique to our design.
>
> # Additional Improvements
> To further strengthen the paper, we have:
> - Included a block efficiency table for temperature=1 sampling in Table 7, 8 in Appendix section G.4.
> - Added a subsection in Methods (Section 3.2) explaining why mask tokens enable multi-token generation, supported by cosine similarity analysis across layers with theoretical justification.
>
> We hope that these revisions and clarifications address the concerns raised and demonstrate the potential impact of our work. Based on these changes, we kindly request the reviewers to reconsider their evaluation.
>
> # References
> [1] Goel, Raghavv, et al. "Direct alignment of draft model for speculative decoding with chat-fine-tuned llms." arXiv preprint arXiv:2403.00858 (2024).
>
> [2] Gagrani, Mukul, et al. "On speculative decoding for multimodal large language models." Proceedings of the IEEE/CVF Conference on Computer Vision and Pattern Recognition. 2024.
>
> [3] Chen, Hao Mark, et al. "Hardware-aware parallel prompt decoding for memory-efficient acceleration of llm inference." arXiv preprint arXiv:2405.18628 (2024).
>
> [4] Samragh, Mohammad, et al. "Your llm knows the future: Uncovering its multi-token prediction potential." arXiv preprint arXiv:2507.11851 (2025).
>
> [5] Eagle integration with vLLM: https://docs.vllm.ai/en/v0.8.1/getting_started/examples/eagle.html

---

### Official Review · Reviewer_uene · 2025-10-31

**Soundness:** 2
**Presentation:** 2
**Contribution:** 2
**Rating:** 2
**Confidence:** 4

**Summary:**

The paper proposes a prompt based probing method for muli-token prediction with LLMs which requires no special training or finetuning. The core of the method is a way to generate special masking tokens that are appended to a prompt in order to stimulate the multi-token prediction.

**Strengths:**

The paper compares different ways to generate the mask tokens. Further, it provides experiments across the LLaMA and Qwen-3 model families, covering performance and speed comparisons.

**Weaknesses:**

While the paper presents an interesting approach, several issues limit its clarity and contribution.

Formatting and Presentation: The overall formatting is not clean, which affects readability. For instance, Table 3 extends beyond the document boundary, making it difficult to interpret. Attention to layout and consistency in formatting would improve the paper’s professionalism.
- Notation Inconsistency: The mathematical and symbolic notations are inconsistent throughout the paper. This inconsistency makes it hard to follow the proposed method and understand the exact mechanisms involved. A clearer and standardized notation scheme would help readers grasp the technical details more effectively.
- Missing Related Work and Comparisons: The paper overlooks several important and directly relevant studies, which weakens the contextualization of the proposed method:
a) arXiv:2505.10518: Introduces a prompt-level multi-token prediction (MTP) approach using trained register tokens instead of heuristic masking, offering a more principled token selection strategy.
b) arXiv:2405.18628: Proposes a method that also employs trained prompt tokens and integrates a dynamic tree structure, providing an alternative perspective on structured token prediction.
c) arXiv:2502.09419: Demonstrates that large language models trained for next-token prediction inherently exhibit multi-token prediction capabilities, an insight that the current paper treats as novel.

**Questions:**

1) How to you compare to the related literature mentioned above
2) In section 3.1 you provide 3 heuristics for mask token injection. Can you provide reasoning why you chose exactly those? Are these the only choices possible?

---

> ### Author Response · Authors · 2025-11-20
> **Response to reviewer uene**
>
> We thank the reviewer for the detailed feedback and have made several improvements to address the concerns:
>
> # General Clarification
> - We would like to reiterate that the core contribution of our paper is not the creation of special mask tokens, but the demonstration that LLMs trained for next-token prediction inherently support multi-token prediction without any additional fine-tuning. Our method is the first to perform training-free speculative decoding by leveraging this implicit behavior.
> - We have simplified notations, improved formatting (including Table 3), and made editorial changes for better readability. We kindly request the reviewer to revisit the strengths of our paper in light of these improvements.
>
> # Formatting and Presentation
> Reviewer Comment: Formatting issues (e.g., Table 3 extending beyond document boundary) and inconsistent notation.
> - Response: We have reformatted Table 3 and standardized notation throughout the paper. Section 4.2 now clearly explains Table 3. Our notation aligns with conventions used in prior work such as BiTA [1].
>
> # Missing Related Work
> Reviewer Comment: Important works were not cited (arXiv:2505.10518, arXiv:2405.18628, arXiv:2502.09419).
> - We have updated Section 5 (Related Work) and Appendix Section B (Extended Related Work) to include a dedicated paragraph on Multi-Token Prediction.
> - Please note that we already cited arXiv:2502.09419, which uses an additional sampler head to refine predictions of learned mask tokens, whereas our method is entirely training-free. This idea is more closely related to EAGLE-style drafting [2].
> - We have added (a) arXiv:2505.10518 and (b) arXiv:2405.18628 in the extended related work in the appendix.
> - Notably, (a) PPD is similar to BiTA [1] but performs worse and uses static pruning, while our method uses dynamic pruning.
> (b) MuTOR updates base model weights and adds register tokens, unlike our approach which requires no weight updates, making it easily deployable across models.
> - We emphasize that training-based methods require significant resources (e.g., BiTA [1] training requires 4 cluster servers with 8xA100 GPUs). While trained methods may outperform online methods, our contribution lies in enabling training-free acceleration without architectural changes. Additionally **all trained methods require retraining as soon as the model changes**. BITA has no results for latest model families of LLaMA3 and Qwen3.
>
> # Question 1: Comparison to Related Literature
> Addressed above with detailed distinctions and resource comparisons.
>
> # Question 2: Why These Three Heuristics for Mask Token Injection?
> - Our intuition was to initialize mask token embeddings within the embedding table bounds, as the model has not seen embeddings outside this range.
> - We include an ablation study in Appendix G.6 (Table 12), where mask tokens are initialized outside the embedding distribution. This results in only a minor drop in average block efficiency, demonstrating robustness to initialization.
> - We also show in Section 3.2 (highlighted in blue for easy reference) that when mask tokens are initialized as the mean of the prompt, cosine similarity between mask and valid token hidden states increases from middle to final layers, which therefore leads to token acceptance as proven in Lemma A.1 in Appendix.
>
> # Additional Improvements
> To further strengthen the paper, we have:
> - Added tokens-per-second (T/S) measurements, showing our method outperforms baselines in Table 1. Our method outperforms other baselines
> - Included a block efficiency table for temperature=1 sampling in Table 7, 8 in Appendix section G.4.
> - Added a subsection in Methods (Section 3.2) explaining why mask tokens enable multi-token generation, supported by cosine similarity analysis across layers with theoretical justification.
> - Performed an ablation on mask token initialization, confirming robustness in Table 12 in Appendix section G.6
>
> We hope that these revisions and clarifications address the concerns raised and demonstrate the potential impact of our work. Based on these changes, we kindly request the reviewers to reconsider their evaluation.
>
> # References
>  [1] Lin, Feng, et al. "BiTA: Bi-directional tuning for lossless acceleration in large language models." Expert Systems with Applications 279 (2025): 127305.
>
>  [2] Li, Yuhui, et al. "Eagle-2: Faster inference of language models with dynamic draft trees." arXiv preprint arXiv:2406.16858 (2024).

---

> > ### Comment · Reviewer_uene · 2025-11-20
> > **Claimed key contribution was published before**
> >
> > Thank you for clarifying on your key contributions and answering my questions.
> >
> > However, I disagree with your statement that this contribution is novel. The paper arXiv:2502.09419, which was published on an ICLR2025 workshop, clearly made this contribution. However, they use a different multi-token prediction scheme based on marginalization. What you claim as novel contribution is explicitly written in the very first two sentences of the abstract and is hard to miss.
> >
> > Let me cite: "We systematically investigate multi-token prediction (MTP) capabilities within LLMs pre-trained for next-token prediction (NTP). We first show that such models inherently possess MTP capabilities via numerical marginalization over intermediate token probabilities, though performance is data-dependent and improves with model scale." Further, many experiments without additional training and MTP heads are given. E.g. Figure 1.
> >
> > Therefore, I would argue the remaining novelty of the paper is your special method to create the mask tokens. Could you comment on that?

---

> > > ### Author Response · Authors · 2025-11-20
> > > **Response on key contribution**
> > >
> > > We thank the reviewer for highlighting arXiv:2502.09419 and for the quick response.
> > >
> > > Our original claim was:
> > > *“LLMs trained for next-token prediction inherently support multi-token prediction without any additional fine-tuning.”*
> > >
> > > The cited paper introduces multi-token prediction by training separate heads and updating base model weights, as noted in their discussion:
> > > *“we explore the challenges of integrating MTP heads into frozen LLMs and find that their hidden layers are strongly specialized for NTP, making adaptation non-trivial.”*
> > >
> > > In contrast, our method is training-free and does not assume independence between generated tokens (as their Eq. (3) does).
> > > This independence assumption simplifies marginalization but ignores inter-token dependencies that are critical for coherent generation. While we acknowledge that the referenced paper successfully demonstrates MTP under this assumption for two tasks, such independence-based modeling can limit generalization and degrade performance for tasks requiring strong token interactions.
> > > Additionally, the referenced paper mentions that MTP performance is dataset and model dependent, whereas our experiments consistently show correct multi-token predictions across diverse evaluation tasks, suggesting a more generalizable phenomenon.
> > >
> > > We will cite this interesting work and clarify our claim to avoid ambiguity:
> > > “LLMs trained for next-token prediction inherently support multi-token prediction without any additional fine-tuning when probed by mask tokens generated on-the-fly.”
> > >
> > > Finally, we emphasize that our work explores MTP in a fundamentally different way—by leveraging probing and dynamic mask token generation rather than architectural modifications or independence-based marginalization.
> > > We hope this clarification highlights the key novelty: a simple, training-free approach versus prior work requiring additional heads, weight updates, and independence assumptions.

---

> > > > ### Comment · Reviewer_uene · 2025-11-21
> > > > **Response on key contribution**
> > > >
> > > > Thank you for your quick response and for your active participation in the discussion.
> > > >
> > > > At this point, I still feel responsible to make you aware that arXiv:2502.09419 clearly addresses both:
> > > > 1) A training free approach which they propose as a strong baseline for training based MTP algorithms in section 2
> > > > 2) A training based approach section 3
> > > >
> > > > Let me quote again:
> > > > "This paper addresses this knowledge gap by investigating MTP adaptation of NTP models. Specifi-
> > > > caly, we make the following contributions:
> > > >
> > > > MTP capabilities of NTP models: We demonstrate that LLMs **trained only for NTP already possess
> > > > MTP capabilities**, i.e. MTP is possible through numeric marginalization over intermediate token
> > > > probabilities. Although this is computationally complex, **it is a strong performance baseline**.
> > > > ..."
> > > >
> > > > **One obvious key result is that training free MTP is possible and that it is a very strong baseline for the training based approaches. Hence, your very strong claims that: 1) all other MTP methods so far require training and that 2) your work is the first that successfully performs training free MTP with vanilla LLMs does not hold**. The insight that training free MTP is possible with vanilla LLMs clearly **was published before and is not novel**!
> > > >
> > > > Please don't get me wrong, your manuscript still has a contribution: A new efficient algorithm for training free MTP. However, in the current manuscript it lacks comparison to the existing approach in terms of 1) speed, 2) generation quality.
> > > >
> > > > I agree that the training free MTP approach proposed in arXiv:2502.09419 is fundamentally different** and most probably much more inefficient. However, I am not sure about **generation quality, which is largely unaddressed in the current manuscript.**

---

> ### Author Response · Authors · 2025-11-21
> **Response on Key Contribution and New Insights on arXiv:2502.09419**
>
> We thank the reviewer for the quick and detailed feedback and highly appreciate the active discussion.
>
> We agree with the reviewer that the referenced paper (arXiv:2502.09419) mentions the statement: “NTP-trained models possess MTP behavior.” However, diving deeper into the referenced paper brings new light mentioned in **Deeper Analysis of the Referenced Paper** section below. In summary, we believe that the referenced paper's training-free MTP claim is only valid till next to next token and after which it becomes intractable (especially for recent models with large vocabulary: LLaMA3 has 126K size).
>
> Additionally, we request the reviewer to kindly check our updated manuscript, where we provide the exact metrics requested—**tokens per second (speed)**—in **Table 1**. Our method outperforms other methods in this metric as well.
> (We also mentioned this in our first response and are reiterating here for clarity.)
>
> Regarding generation quality, please note that all generations in our method are lossless, as we perform sample matching, i.e., a token is only accepted if it exactly matches the predicted token. Details of this mechanism are provided in **Section 2 (Background)**. Note that the baseline methods (lookahead decoding) also perform sample matching.
>
> # Deeper Analysis of the Referenced Paper
> - We respectfully request the reviewer to examine **Equation (6)** in the referenced paper. To elicit MTP behavior, the authors iterate over all tokens in the vocabulary and pass each through the model, requiring more than a single forward pass to produce a block (K) of tokens. In contrast, **our proposal predicts K next tokens in parallel in a single forward pass**. We believe this is a key distinction between **true multi-token prediction** and **intractable multi-token prediction**.
>
> - For reference, the equation (6) for predicting the next-to-next token $x_{t+2}$:
>   - $p(x_{t+2}|X_{\leq t}; \theta) = \Sigma_{y\in V}p(x_{t+2}|X_{\leq t}, y;\theta)p(y|X_{\leq t};\theta)$
>   - Here $ p(x_{t+2}|X_{\leq t}, y;\theta)$ requires forward pass for each $y$ (over vocabulary set or a subset). This becomes intractable for predicting $x_{t+3}$:
>   - $p(x_{t+3}|X_{\leq t}; \theta) = \Sigma_{y_1\in V,y_2\in V}p(x_{t+2}|X_{\leq t}, y_2, y_1;\theta)p(y_2|X_{\leq t}, y_1;\theta)p(y_1|X_{\leq t};\theta)$
>   - Here we need to get probability for all combinations of $y_1, y_2$ which cannot be run in parallel (compute bound).
>   - Additionally for finding a smaller set of tokens, we would still require forward pass of the model to get $p(y_1|X_{\leq t};\theta)$ or $p(y_2|X_{\leq t}, y_1;\theta)$. Therefore, beyond next to next token, we don't see how this method is predicting good tokens. In our method we show that using two mask tokens (predicting $x_{t+2}, x_{t+3}$) is beneficial tasks like coding, math/reasoning.
>
> - While we acknowledge the relevance of the referenced work, we note that it was presented as a workshop paper and does not provide complete experimental results for the training-free baseline (e.g., accuracy metrics) in Table 1 even though as the reviewer pointed out we quote "Although this is computationally complex, **it is a strong performance baseline**".
> - Our intention is not to diminish its contribution but to highlight that the lack of detailed evaluation makes direct comparison and claims presented in the paper challenging. We hope the reviewer considers this context when assessing the novelty and completeness of our work.
>
> # Summary of Our Request
> We kindly request the reviewer to re-evaluate our paper considering:
> - The referenced paper’s (arXiv:2502.09419) approach is intractable beyond 2 token MTP, requiring additional forward passes.
> - Our updated manuscript demonstrates that our proposed method outperforms other baselines in tokens per second (speed).
> - Our method guarantees lossless generation through sample matching.
> - New insights on decoder layers are adding relevant information to mask token states, which when leads to high cosine-similarity between mask and valid token state leads to acceptance otherwise rejection.
>
> We hope these clarifications and updates address the concerns raised and help in reassessing the novelty and contribution of our work.

---

> > ### Comment · Reviewer_uene · 2025-11-25
> > **Thank you for the discussion**
> >
> > Thank you for the discussion. I agree with you on all points of your summary and
> > raise my score accordingly.

---

> ### Author Response · Authors · 2025-11-25
> **Thanks for raising score and request to review new stronger theoretical insight**
>
> We really thank the reviewer for raising the score and for active discussion.
>
> We will be grateful if the reviewer can also check the below where we mention our new theoretical insight. We hope that this can help further strengthen our contribution. We have updated the abstract and contributions part of Introduction accordingly.
>
> **We have strengthened the paper by replacing our earlier proof with a more rigorous and general theoretical result, supported by empirical evidence. We request all reviewers to check Section 3.2 in main body of paper**
>
> Our method is based on the observation that decoder layers progressively enrich the mask token representation, aligning its hidden state with that of valid tokens. This alignment is crucial because the LM head computes logits via inner products between the final hidden state and vocabulary embeddings $w_{r} \in R^{d}$. A higher inner product with the correct token embedding increases its logit, making it more likely to appear in the Top-$K$ candidate set.
>
> To quantify this phenomenon, we **track the evolution of cosine similarity between the mask token and the next-true token across layers**. Empirically, accepted tokens show a steady increase in similarity after layer 15 (up to ~0.45), while rejected tokens plateau near ~0.35 (see Figure 2 in updated manuscript). This divergence strongly correlates with acceptance.
>
> # New Theoretical Contribution
> We formalize this intuition with a lemma: Lemma 3.1 in Section 3.2 (please check Appendix A for detailed proof) that establishes a sufficient condition for predicted token acceptance: if the **cosine similarity between the mask token and the next-true token exceeds a threshold $\delta^*$, then the next-next true token (under greedy sampling) is guaranteed to be in the Top-$K$ predictions of the mask token**. This result provides a principled link between representation alignment and prediction accuracy, improving the rigor and interpretability of our approach.
>
> # Impact
> This addition significantly enhances the paper’s quality by moving beyond empirical performance gains to explain why the method works, offering theoretical guarantees and actionable insights for future research on speculative decoding and representation alignment.

---

### Author Response · Authors · 2025-11-24
**New Rigorous Theoretical and Empirical Justification Added (all reviewers)**

**We have strengthened the paper by replacing our earlier proof with a more rigorous and general theoretical result, supported by empirical evidence. We request all reviewers to check Section 3.2 in main body of paper**

Our method is based on the observation that decoder layers progressively enrich the mask token representation, aligning its hidden state with that of valid tokens. This alignment is crucial because the LM head computes logits via inner products between the final hidden state and vocabulary embeddings $w_{r} \in R^{d}$. A higher inner product with the correct token embedding increases its logit, making it more likely to appear in the Top-$K$ candidate set.

To quantify this phenomenon, we **track the evolution of cosine similarity between the mask token and the next-true token across layers**. Empirically, accepted tokens show a steady increase in similarity after layer 15 (up to ~0.45), while rejected tokens plateau near ~0.35 (see Figure 2 in updated manuscript). This divergence strongly correlates with acceptance.

# New Theoretical Contribution
We formalize this intuition with a lemma: Lemma 3.1 in Section 3.2 (please check Appendix A for detailed proof) that establishes a sufficient condition for predicted token acceptance: if the **cosine similarity between the mask token and the next-true token exceeds a threshold $\delta^*$, then the next-next true token (under greedy sampling) is guaranteed to be in the Top-$K$ predictions of the mask token**. This result provides a principled link between representation alignment and prediction accuracy, improving the rigor and interpretability of our approach.

# Impact
This addition significantly enhances the paper’s quality by moving beyond empirical performance gains to explain why the method works, offering theoretical guarantees and actionable insights for future research on speculative decoding and representation alignment.

---

### Author Response · Authors · 2025-12-01
**Dear Area Chair summary of rebuttals and discussions (1/2)**

We sincerely appreciate your time and the extra effort you’ve provided due to the unfortunate incident. Below is a detailed summary of our responses and discussions with the reviewers, which we believe address most of their concerns. All major changes in the manuscript are highlighted in blue for ease of review.
We use the following notations: Ri (reviewer i), W for weakness, and Q for question.

# Reviewer 1
- **R1W1 (Formatting and presentation)**: Resolved. Table 3 now fits within page width, and notations are consistent with prior works (e.g., BiTA [1]).
- **R1W2 (Missing related work)**: Resolved. Please see Section 5 and Appendix B for updated related work. Missing citations have been added, and their limitations are discussed.
- **R1Q1 (Comparison with training-based methods)**: Resolved. Training-based methods require significant compute (e.g., BiTA-based acceleration needs 4 cluster servers with 8×A100 GPUs) and retraining for each new model, unlike our training-free, model-agnostic approach.
- **R1Q2 (Heuristics for mask token injection)**: Resolved. Mask tokens initialized within the embedding distribution (mean of prompt embeddings) perform best. Ablation in Appendix G.6 (Table 12) shows initializing outside the distribution degrades performance. Section 3.2 includes new results on increased cosine similarity when using the mean of prompt embeddings.
- **R1Q3 (First training-free multi-token prediction)**: Resolved. Our work is the first true training-free multi-token prediction, unlike [2], which uses impractical MTP. Please refer to our discussion with the reviewer.
- **R1Q4 (Tokens per second and temperature=1 results)**: Resolved. Tokens per second are added in Table 1. Sampling results at temperature=1 are in Appendix G.4 (Tables 7, 8), showing generations remain lossless due to sample matching.
- **Note**: We greatly appreciate R1 increasing their score after our discussion.

# Reviewer 2
- **R2W1 (Clarification on BE, BC, and tokens per second)**: Resolved. BE = 1 + MAT (mean accepted tokens), as noted in Section 4 (lines 350–351). Table 1 includes tokens per second. Our method outperforms other training-free methods in both MAT and tokens per second. We also introduce BC (block complexity), defined as the number of input tokens processed in a single forward pass.
- **R2W2 (Limited model coverage, missing Qwen)**: Resolved. Qwen is included, along with four model sizes: 3B, 7B, 8B, and 32B.
- **R2W3 (Onboarding vLLM complexity)**: Resolved. Baseline methods also use tree-attention masks. Moreover, EAGLE-2 (a training-based speculative decoding method using tree-attention masks) is already integrated with vLLM, implying our method should be relatively easy to integrate. Please check [3] for more details.

# Reviewer 3
- **R3W1 (Practical deployment with vLLM)**: Resolved. Our primary research question is whether modern LLMs trained on next-token prediction can predict multiple tokens without training and whether this can accelerate inference—a fundamental capability question rather than an engineering optimization. Tree-attention-based methods like EAGLE-2 are already integrated with vLLM, so our method and baselines should integrate easily. Please check [3] for more details
- **R3W2 (Generalization beyond greedy decoding)**: Resolved. Sampling with temperature=1 is added in Appendix G.4 (Tables 7, 8). Our method achieves the highest BE in this setup as well. Since our approach generates a tree of draft tokens, it works well even for sampling-based generation.
- **R3Q1 (Metric beyond BE)**: Resolved. Tokens per second are added in Table 1.


[1] Lin, Feng, et al. "BiTA: Bi-directional tuning for lossless acceleration in large language models." Expert Systems with Applications 279 (2025): 127305.

[2] Mehra, Somesh, Javier Alonso Garcia, and Lukas Mauch. "On multi-token prediction for efficient LLM inference." arXiv preprint arXiv:2502.09419 (2025).

[3] Eagle integration with vLLM: https://docs.vllm.ai/en/v0.8.1/getting_started/examples/eagle.html

---

> ### Author Response · Authors · 2025-12-01
> **Dear Area Chair summary of rebuttals and discussions (2/2)**
>
> # Reviewer 4
> - **R4W1 (Discussion on masked diffusion LLMs)**: Resolved. Added in future work (Appendix C). Our focus is on autoregressive LLMs trained for single-token prediction, unlike diffusion LLMs trained for multi-token prediction per forward pass.
> - **R4W2 (Why probing works)**: Resolved. A new Section 3.2 explains theoretical and empirical reasons. Ablations in Appendix G.9 (Table 12) show performance drops when mask tokens are initialized far from the embedding distribution. Intuitively, mask tokens act as noisy next-true tokens, and each decoder layer aligns them closer to the true token state.
> - **R4W3 (Wall-clock speed-up)**: Resolved. Tokens per second (equivalent to wall-clock speed-up) are added in Table 1; our method achieves the highest speed-up.
> - **R4Q1 (Why only greedy decoding)**: Resolved. Most speculative decoding methods use greedy decoding. We added sampling-based results (temperature=1) in Appendix G.4 (Tables 7, 8), where our method still outperforms baselines.
> - **R4Q2 (Comparison with training-based methods)**: Resolved. Training-based methods require significant compute, retraining, and high-quality datasets for each new model, even though they outperform training-free methods. Our approach is plug-and-play across models without retraining—a key practical advantage.
> - **R4Q3 (Why shift isn’t a problem)**: Resolved. Section 3.2 explains that mask tokens predict next-next tokens with corresponding position indices. For example, given $n$ past tokens, the last true token ($x_n$) predicts $x_{n+1}$, and mask tokens predict potential (draft) $x_{n+2},…,x_{n+k}$.
> - **R4Q4 (Clarification on R4W2)**: Resolved. Our intuition is that mask token hidden state resembles a noisy true hidden state: $h_{m,n+1}^l=h_{x,n+1}^l+\epsilon^l$ for any layer $l$ and future positions $n+i$. Noise decreases as the state passes through decoder layers. Our **new figure 2** shows that similarity increases during token acceptance which implies noise could be decreasing.
> - **R4Q5 (Methods a), b), c) are natural)**: Resolved. These methods explore different initialization strategies for mask tokens while keeping them within the embedding distribution. Table 5 shows option (c)—initializing with the mean of prompt embeddings—performs best. Performance drops as initialization moves away from the embedding distribution as shown in Appendix G.9 Table 12.
>
> # General Updates
> 1. Added new theoretical (**Lemma 3.1**) and empirical insights into why mask tokens enable correct next-next token predictions. This is now listed as a contribution.
> 2. Post-discussion with R1, we can claim our work is the first to enable true training-free multi-token prediction by leveraging the model’s internal representation.
> 3. New tokens-per-second results demonstrate wall-clock speed-up.
> 4. Sampling-based decoding results confirm our method remains the best.
> 5. New results on mask token initialization within and outside embedding table distribution.
>
> We hope these detailed updates, new results, and clarifications resolve all concerns. We truly believe our research addresses a fundamental question:
>
> **“Can autoregressive LLMs predict multiple tokens in a training-free manner?”**
>
> —a question that has not been explored rigorously before.
>
> Thank you again for your time and effort in understanding our work and rebuttals.

---

### Meta-Review · Area_Chair_AeLp · 2026-01-07

**Summary:**

Reviewers raised substantial concerns regarding the novelty and technical soundness of the proposed training-free multi-token prediction method. In particular, several reviewers questioned whether the probing mechanism and mask-token construction meaningfully differ from or improve upon existing speculative and lookahead decoding techniques. Additional concerns focused on the limited scope of experimental evaluation, unclear ablation of key design choices, and whether the reported efficiency gains are robust across tasks, models, and decoding settings.

**Reviewer Concerns:**

The rebuttal clarified parts of the method design and provided additional intuition for the probing mechanism, addressing some misunderstandings about how mask tokens are constructed and used during inference. These explanations improved clarity but primarily at the level of presentation. Core concerns regarding the lack of clear novelty, insufficient empirical validation, and the absence of a strong theoretical justification remain largely outstanding. Overall, the rebuttal did not convincingly demonstrate that the proposed approach constitutes a principled or broadly impactful advance over existing training-free decoding methods.

**Reviewer Scores:**

2244. If reviewers had been able to fully participate in post-rebuttal discussion, minor score increases reflecting improved clarity might have occurred. However, given the severity of the remaining concerns, it is unlikely that such changes would have altered the overall assessment or the final recommendation.

---

### Decision · Program_Chairs · 2026-01-26

Reject